# A flexible artificial intrinsic-synaptic tactile sensory organ

Yu Rim Lee[1], Tran Quang Trung[1], Byeong-Ung Hwang[1] & Nae-Eung Lee [1,2,3,4,5]✉

Imbuing bio-inspired sensory devices with intelligent functions of human sensory organs has been limited by challenges in emulating the preprocessing abilities of sensory organs such as reception, filtering, adaptation, and sensory memory at the device level itself. Merkel cells, which is a part of tactile sensory organs, form synapse-like connections with afferent neuron terminals referred to as Merkel cell-neurite complexes. Here, inspired by structure and intelligent functions of Merkel cell-neurite complexes, we report a flexible, artificial, intrinsic-synaptic tactile sensory organ that mimics synapse-like connections using an organic synaptic transistor with ferroelectric nanocomposite gate dielectric of barium titanate nanoparticles and poly(vinylidene fluoride-trifluoroethylene). Modulation of the post-synaptic current of the device induced by ferroelectric dipole switching due to triboelectric-capacitive coupling under finger touch allowed reception and slow adaptation. Modulation of synaptic weight by varying the nanocomposite composition of gate dielectric layer enabled tuning of filtering and sensory memory functions.

[1] School of Advanced Materials Science & Engineering, Sungkyunkwan University, 2066 Seobu-ro, Jangan-gu, Suwon-si, Gyunggi-do 16419, Korea. [2] SKKU Advanced Institute of Nano Technology (SAINT), Sungkyunkwan University, 2066 Seobu-ro, Jangan-gu, Suwon-si, Gyunggi-do 16419, Korea. [3] Samsung Advanced Institute for Health Sciences and Technology (SAIHST), Sungkyunkwan University, 2066 Seobu-ro, Jangan-gu, Suwon-si, Gyunggi-do 16419, Korea. [4] Institute of Quantum Biophysics (IQB), Sungkyunkwan University, 2066 Seobu-ro, Jangan-gu, Suwon-si, Gyunggi-do 16419, Korea. [5] Biomedical Institute for Convergence at SKKU (BICS), Sungkyunkwan University, 2066 Seobu-ro, Jangan-gu, Suwon-si, Gyunggi-do 16419, Korea. ✉email: nelee@skku.edu

Sensory organs enable animals to gather information for perception and to have ability for lives, such as conducting skilled movements and seeking protection from hazardous situations. Sensory organs consist of stimuli-sensitive cells such as photoreceptors for vision[1,2], chemoreceptors for olfaction and gustation[3], and mechanoreceptors for audition[4], all of which have synapse-like connections to afferent neurons. Synapses in the brain allow information processing in parallel with high energy efficiency, and there is some evidence that sensory organs with synapse-like connections with neurons also preprocess information via intelligent functions including adaptation, filtering, amplification, and memory before transmitting the information to the brain[5–10]. Merkel cells are cutaneous mechanosensitive cells that form synapses with afferent neurons, and these complexes are referred to as Merkel cell-neurite complexes (MCNCs)[11]. There have been some reports on artificial tactile sensors mimicking mechanoreceptors with slowly adapting (SA) firing analysis[12,13]. Even though much effort has been made to mimic mechanoreceptors in the human body[14–18], emulation of their intelligent functions to extend sensory reception has not been widely successful due to difficulty in adding synaptic connection between receptors and neuron terminal at the device level. Several bio-inspired tactile reception systems have recently been developed. For example, an artificial sensory nerve was developed by connecting a piezoresistive pressure sensor, a ring oscillator generating action potential, and an ion-gel-based synaptic transistor[19]. A tactile perceptual learning system was realized by integration of a piezoresistive touch sensor, an ionic cable, and an ion-gel-based synaptic transistor[20]. These bio-inspired tactile reception systems were developed by integrating discrete synapse devices and sensors mimicking the reception function of mechanoreceptors such as Meissner's or Pacinian corpuscles. However, the sensors themselves are not intelligent, in contrast to mechanoreceptors in the human body, and complicated fabrication processes are required to generate the discrete components of the systems.

Herein, we demonstrate a flexible, artificial, intrinsic-synaptic tactile sensory organ (AiS-TSO) that mimics the synapse-like connections of MCNCs, conferring intrinsic-synaptic properties to the unit sensor for conducting further intelligent work. This AiS-TSO is based on a flexible ferroelectric organic field-effect transistor (Fe-OFET) gated by a triboelectric-capacitive coupling effect. Touch stimulation induces alignment of dipoles in the ferroelectric gate dielectric by triboelectric-capacitive coupling effect which causes the post-synaptic current signal to be modulated, thereby allowing tactile information to be imparted to the signal in a self-energy transducing manner. The synaptic function of the device enables the output signal to be pre-processed though the multiple functions of SA, filtering and memory in a self-energy transducer manner. Tunability of SW in the AiS-TSO was achieved by varying the composition of the nanocomposite ferroelectric layer of barium titanate nanoparticles (BT NPs) and poly(vinyledenedifluoride–trifluoroethylene) (P(VDF-TrFE)). We also emulated sensory memory in a $2 \times 2$ sensor array by recognizing the number and order of touch without additional signal processing after all stimuli ceased. We believe that the concept of AiS-TSO represents a new paradigm in sensor technology for artificial intelligence (AI) and autonomous systems requiring real-time decisions and highly energy-efficient sensing.

## Results

**Mimicking human tactile sensory organ.** Even though a well-integrated theory of the reception mechanism of MCNCs is lacking[21–23], influx of $Ca^{2+}$ ions through mechanoresponsive $Ca^{2+}$ gating ion channels (Piezo-2) in Merkel cells increases membrane potential and release of neurotransmitters, resulting in SA sensations with modulation of SW (Fig. 1a)[23,24]. By emulating the synapse-like connections of a sensory organ, biomimetic tactile sensors with information preprocessing ability at the device level could be used as intrinsically intelligent devices in future intelligent systems. For an explanation of the analogy between MCNCs and AiS-TSO, see Supplementary Table 1 and Supplementary Note 1. A schematic illustration of an AiS-TSO mimicking the structure and functions of an MCNC in the human body is provided in Fig. 1. To mimic the stimuli reception of Piezo-2 channels in Merkel cells that convert mechanical energy to potential, we exploited a triboelectric-capacitive coupling effect on the receptive part (substrate) of the device. Touch cause triboelectric charge pumping to occur from the finger to the receptive part, similar to the inflow of $Ca^{2+}$ ions through the Piezo-2 channel. The tribo-capacitance of the receptive part, which increases in response to pumped electrons, induces a polarization change of the ferroelectric layer, which is analogous to neurotransmitter release from the Merkel cell. The change in orientation of permanent dipoles in the ferroelectric gate dielectric modulates the channel conductivity of an Fe-OFET, acting as SW control. The drain current of the device is equivalent to the post-synaptic current (PSC, $I_{PSC}$), which is controlled by the generated tribo-capacitive potential (receptor potential, $V_{rec}$).

**Synaptic properties of Fe-OFET.** We first fabricated and characterized an Fe-OFET device with BT NP(20 wt%)/P(VDF-TrFE) nanocomposite gate dielectric (thickness of 0.6 μm) and Ni gate electrode on polyimide (PI) substrate using pentacene as organic semiconductor channel to investigate the synaptic properties of the AiS-TSO (Fig. 2a). Notably, the tribo-capacitive potential induced by touch in the tactile reception experiment was replaced with biasing of the gate electrode (also represented as $V_{rec}$) for full characterization of the synaptic characteristics. Details of the fabrication process of the Fe-OFET are provided in the Methods section. Fundamental output characteristics, transfer characteristics and gate leakage current of the Fe-OFET are shown in Supplementary Fig. 1.

The schematic in Fig. 2b illustrates the principles of synaptic functions. When we applied a negative $V_{rec}$ to the gate electrode, dipoles in the ferroelectric layer aligned in a downward direction, resulting in increased accumulation of hole carriers in the p-type channel and, in turn, an increase in drain current, i.e., PSC (excitatory PSC). In contrast, when we applied positive $V_{rec}$ to the gate electrode, dipoles in the ferroelectric layer changed their orientation to the opposite direction, resulting in a decrease in PSC (inhibitory PSC). When $V_{rec}$ was removed, polarization in the same direction as with $V_{rec}$ was generated or not depending on the nature of $V_{rec}$. The state change of polarization can be considered analogous to the memory process while restoration of channel conductance can be considered analogous to the forgetting process. To characterize the synaptic properties of the Fe-OFET, we defined the synaptic weight (SW) as the current change ratio of initial PSC ($I_{PSC,i}$) to the relaxed current after 15 s ($I_{sw}$), which we expressed as ($I_{sw} – I_{PSC,i}$)/$I_{PSC,i}$ (Fig. 2c). Depending on the resulting varied retention time of polarization, we classified SW as short-term or long-term plasticity (STP/LTP). To obtain STP synaptic properties, we applied a $V_{rec}$ of −10 V with a frequency of 0.1 Hz and pulse width of 0.5 s to the device; the peak PSC ($I_{PSC}$) value was approximately −6 nA, and it decayed stably (~−2 nA) within ~5 s (Fig. 2d). In contrast, LTP was observed at a higher frequency of 1.42 Hz with the same pulse width of 0.5 s (Fig. 2e). Synaptic functions of STP and LTP were effectively realized in low (0.1–1.42 Hz, pulse width of 0.5 s) and high (1–8 Hz, pulse width of 0.1 s) frequency ranges at amplitude

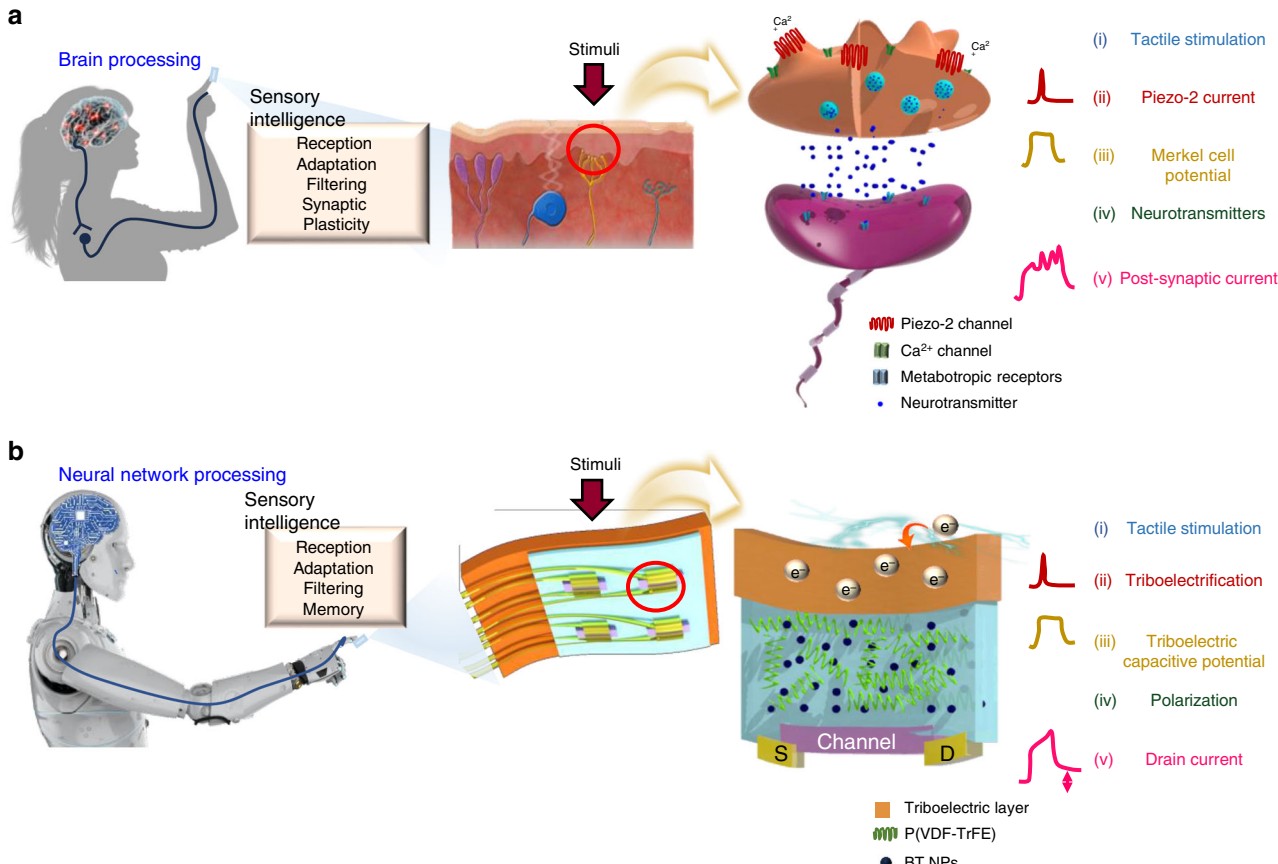

**Fig. 1 Merkel cell-neurite complex and artificial, intrinsic-intelligent tactile sensory organ (AiS-TSO). a** Sensation and processing of stimuli by a sensory organ comprising a Merkel cell and neuron. The MCNC comprises the slowly adapting (SA) sensing mechanism of a Merkel cell and a synapse-like connection with an afferent neuron. Pressure induces influx of $Ca^{2+}$ through the Piezo-2 channel into the Merkel cell, increasing membrane potential. The increase in membrane potential is prolonged because a Merkel cell has a high membrane resistance, which contributes to SA synaptic firing through release of neurotransmitters. SA firing occurs in response to tactile information and is processed by synaptic plasticity. **b** Reception by a flexible AiS-TSO occurs in a self-energy transducing way by triboelectrification during finger touch. Triboelectrification induces tribo-charge pumping in the receptive part, which generates a triboelectric capacitive potential. Drain current is modulated with aligned dipoles in the BT NPs/P(VDF-TrFE) nanocomposite by generated triboelectric capacitive potential, resulting in a SA signal with synaptic properties. The same colors in **a**, **b** indicate corresponding functions of the MCNC and AiS-TSO device.

of $-10$ V for potentiation (Supplementary Fig. 2 and Supplementary Note 2). We fabricated an OFET using polyvinylpyrrolidone (PVP) to demonstrate that the SW generation mechanism of Fe-FET is due to partial polarization. SW in the OFET with PVP was negligible compared to that in the Fe-OFET with BT NP (20 wt%)/P(VDF-TrFE). Other mechanism such as charge trapping in organic semiconductor[25,26] may not be not be enough to induce SW in the OFET with PVP (Supplementary Fig. 3 and Supplementary Note 3). Therefore, generation of SW in the Fe-OFET is mainly attributed to ferroelectric partial polarization switching.

Another parameter of SW is the paired pulse ratio (PPR), which is defined as the ratio of two peak currents (A2/A1), as shown in Fig. 2c. PPR is representative of STP, which is the subsequent output enhanced by previous stimuli[27–29]. To emulate the PPR of a biological synapse, we applied two successive $V_{rec}$ values with different pulse widths (Supplementary Fig. 4). The PPR value decreased as the time interval between two successive pulses ($\Delta t_{rec}$) at both pulse widths increased.

Release of neurotransmitters varies depending on membrane potential, which is related to changes in SW. We demonstrated spike amplitude-dependent plasticity (SADP) by increasing the amplitude of $V_{rec}$ ($-1, -3, -5, -7, -10, -15,$ and $-20$ V) with the same pulse width of 0.5 s (Fig. 2f). The SW value of 9.95 at a

$V_{rec}$ of $-20$ V was larger by a factor of 17 than that at a $V_{rec}$ of $-7$ V (SW = 0.56), while the SW at a $V_{rec}$ of $-1$ V of was negative (SW = $-0.30$). This means that there was no change in polarization for $V_{rec} = -1$ and $-3$ V (SW = $-0.01$) or $-5$ V (SW = $-0.05$), indicating STP properties. Furthermore, characteristics of spike number-dependent plasticity (SNDP) were obtained by increasing the number of pulses while keeping the amplitude and pulse width of $V_{rec}$ constant at $-10$ V and 0.5 s, respectively. The value of $I_{PSC}$ gradually increased as the spiking number increased from 1 to 100 (Fig. 2g). Further investigation of SW indicated that SW values increased as the frequency (spike rate-dependent plasticity, SRDP), duration time (spike duration time-dependent plasticity, SDDP), and number of pulses increased (Supplementary Fig. 5). In addition, we demonstrated depression of SW under the same conditions except positive $V_{rec}$ amplitude (Supplementary Fig. 5). Also, we measured the $I_{PSC}$ of Fe-OFET by applying $V_{rec}$ pulses with their amplitude and number consecutively increased or decreased after full recovery to confirm repeatability in the synaptic characteristics of Fe-OFET. The $I_{PSC}$ values measured were almost similar when measured with increasing or decreasing amplitude or number of $V_{rec}$ pulses (Supplementary Fig. 6 and Supplementary Note 4).

Summing up the phenomena presented in Fig. 2, the state change of polarization controlling the synaptic properties was

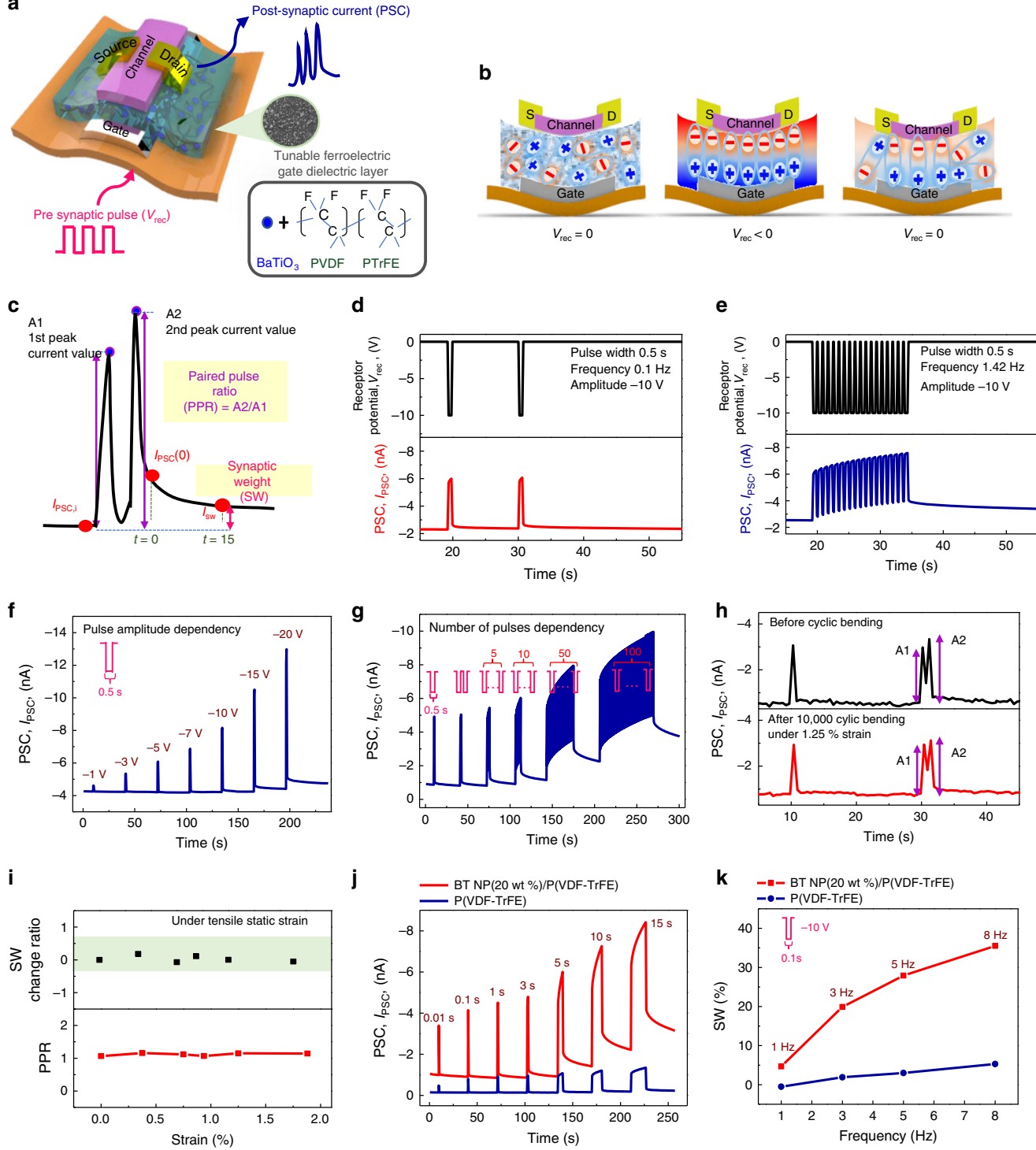

**Fig. 2 Analysis of synaptic properties of Fe-OFET. a** Sketch of device structure. Pre-synaptic pulse corresponds to an electrical gate bias voltage, $V_{rec}$. Gate dielectric layer consists of BT NPs/P(VDF-TrFE) ferroelectric material. **b** Mechanism by which the post-synaptic current (PSC) is generated. Schematic explaining the alignment of ferroelectric dipoles before applying $V_{rec}$ (left), during application of a negative $V_{rec}$ (middle), and after removal of the $V_{rec}$ (right). **c** Schematic explanation of synaptic weight (SW) and paired pulse ratio (PPR). **d, e** Short-term potentiation (frequency of 0.1 Hz, **d**) and long-term potentiation (frequency of 1.42 Hz, **e**) were triggered by applying −10 V $V_{rec}$ pulses. PSC dependence on **f** $V_{rec}$ amplitude (pulse width, 500 ms). **g** Number of $V_{rec}$ pulses (amplitude of −10 V and pulse width of 500 ms). **h** PSC before (black) and after (red) a cyclic bending test. PSC in response to a pulse and successive pulses of −10 V $V_{rec}$ with a pulse width of 500 ms (pulse interval of successive identical pulses, $\Delta t_{rec} = 500$ ms). **i** Variations in SW change ratio and PPR under different static tensile strain. PSC of devices with BT NP(20 wt%)/P(VDF-TrFE) (red) and P(VDF-TrFE) only (blue) (**j**), as a function of $V_{rec}$ pulse duration. (amplitude of $V_{rec}$, −10 V) (**k**), at different frequencies of $V_{rec}$ (amplitude of −10 V).

dependent on the nature of the applied $V_{rec}$. The larger, longer, and more repetitive was the applied $V_{rec}$, the larger was the polarization generated. Generation of SW in the AiS-TSO with ferroelectric nanocomposite can be explained by partial polarization switching at low electric field, similarly to previous investigations on polarization switching dynamics of minor loops in ferroelectric materials[30–35]. We investigated the partial polarization in minor loops of BT NP(20 wt%)/P(VDF-TrFE) thin film with measurement of P–E curves (Supplementary Fig. 7 and Supplementary Note 5). We usually use the pulse amplitude of ±10 V using minor loops of ferroelectric layer rather than the saturation loop since we could get synaptic properties which can be used for sensory memory[8,23,24] by controlling the partial polarization switching in ferroelectric gate dielectric layer and so control the STP and LTP properties by varying the duration, number and frequency of pulses (Fig. 2). Of course, when we compare the memory retention, the retention time was much smaller (~68 min) with −10 V of $V_{rec}$ pulses applied than that with the pulse amplitude of −30 V was applied (~1814 min) (Supplementary Fig. 8 and Supplementary Note 6). The results are consistent with the concerns about using minor-loop of ferroelectric materials addressed in the previous reports[33,36–39]. However, high linearity of PSC was advantageous when we use the −10 V than −30 V as $V_{rec}$ pulse amplitude as shown in Supplementary Fig. 8.

Since we used partial polarization of ferroelectric gate insulating layer, we did not conduct poling process. In case of generating SW by using the partial polarization, poling process on the device which makes the saturated polarization switching resulting in formation the internal field in gate insulating layer was disadvantageous for generation of SW with both potentiation and depression. (Supplementary Fig. 9 and Supplementary Note 7).

After $V_{rec}$ was turned off, the increased polarization slowed the recovery of the PSC and generated SW. The stronger was the polarization, the slower was the decay of the PSC due to a larger SW. The decay (retention) process was analyzed quantitatively using the exponential decay function shown in Eq. (1):

$$I_{PSC}(t) = I_{PSCi} + \left(I_{PSC}(0) - I_{PSC,i}\right)\exp(-t/\tau) \qquad (1)$$

Here, $t$ is the time after applying $V_{rec}$, $I_{PSC}(t)$ is $I_{PSC}$ at time $t$, and $I_{PSC}(0)$ the drain current at $t = 0$. $\tau$ refers to the decay time constant that depends on the frequency, duration, and number of $V_{rec}$ pulses. As the frequency, number of pulses, and duration increased, $\tau$ values increased and were larger for the device with BT NP(20 wt%)/P(VDF-TrFE) than for that with P(VDF-TrFE) only (Supplementary Fig. 10). These results indicate that repetitive, frequent, and longer stimuli can be memorized with longer decay time constants, similar to biological consolidation processes in which sensory memory transitions to long-term memory (LTM) due to SW[40].

Mechanical flexibility is another required characteristic for soft and intelligent bio-mimic electronics. Therefore, evaluation of synaptic functions and endurance was carried out under static and cyclic bending tests. Comparison of $I_{PSC}$ values after 100,000 bending cycles at 1.25% tensile strain showed that variation in the PPR value of the device after cyclic bending was <0.1 (Fig. 2h). Relative variations in PPR, SW, $\Delta PPR/PPR_0$, and $\Delta SW/SW_0$, where $PPR_0$ and $SW_0$ indicate PPR and SW under no mechanical strain, respectively, were also negligibly small under static tensile strain up to 1.88% (Fig. 2i). Additional results from flexibility tests indicated that variation in synaptic characteristics was also negligible under compressive strain (Supplementary Fig. 11).

In addition, we were able to tune the SW of the device by controlling the fraction of BT NPs in the nanocomposite gate

dielectric. We found that the BT NP(20 wt%)/P(VDF-TrFE) nanocomposite has a smaller coercive field (48.83 MV/m) than that of pure P(VDF-TrFE) (88.2 MV/m), which implies easier polarization switching in the nanocomposite than in P(VDF-TrFE). Nevertheless, the coercive field of our P(VDF-TrFE) layer is significantly higher than reported elsewhere[41,42]. Therefore, the Fe-OFET device with BT NP(20 wt%)/P(VDF-TrFE) nanocomposite had a higher hysteresis than the device with no BT NP (Supplementary Fig. 1). Because the dielectric constant of BT NPs/P(VDF-TrFE) nanocomposites is higher than that of P(VDF-TrFE), the PSC of the Fe-OFET with the ferroelectric nanocomposite was greatly enhanced due to the increased dielectric constant (Fig. 2j). In addition, the device with 20 wt% BT NP showed a larger rise in PSC (~3.7 nA) than that without BT NPs (~0.8 nA), indicating an increase in polarization ($V_{rec}$ amplitude of −10 V and pulse width of 1 s). The SW data in Fig. 2k show that the SW value at different frequencies depends on the fraction of BT NPs. When $V_{rec}$ at a frequency of 1 Hz (amplitude of −10 V and pulse width of 0.1 s) was applied, the SWs of the devices with nanocomposite (20 wt% BT NP) and only P(VDF-TrFE) were 4.67 and −0.52, respectively. The SW of the nanocomposite device at a $V_{rec}$ of 8 Hz was 35.50, which was seven times higher compared with that of the P(VDF-TrFE) device. Investigation of synaptic characteristics by comparing the devices with nanocomposite (20 wt% BT NP) or P(VDF-TrFE) gate dielectrics showed that increasing duration, frequency, and number of stimuli increased the SW value (Supplementary Fig. 5). Furthermore, the degree of change in SW could be tuned by varying the fraction of BT NPs in the nanocomposite ferroelectric layer. Comparison of $\tau$ for the devices with nanocomposite (20 wt% BT NP) or P(VDF-TrFE) gate dielectrics revealed that the nanocomposite device had a longer polarization retention time (Supplementary Fig. 10). These results indicate that tunability of the synaptic properties and SW of the Fe-OFET used in our AiS-TSO confer it with intelligent properties as well as controllability of its sensory functions. One of factors affecting SW is the device scaling which includes changes in the thickness of ferroelectric layer and channel length or width[43–48] in Fe-FET. Decrease in the channel length of our Fe-OFET enhanced $I_{PSC}$ (Supplementary Fig. 12). However, increase in the channel length enhanced SW due to larger retention time which might be related to slower polarization switching[44,46,49].

**Synaptic functions of AiS-TSO.** There is still active research about structural and phenomenological observations on MCNCs[11,23,24,50] including the relationship between the number and spatial density of MCNCs and SA perception[51]. Although the exact mechanism of SA firing in MCNCs has not been discovered, it is obvious that there are complex interactions between mechanosensitive Piezo-2 channels, cell membrane potentials, and synergistic synapses of MCNCs that allow Merkel cells to initiate Aβ afferent pulses to encode tactile information[9,22,52]. This suggests that the functions of reception and information preprocessing in our body are not independent of each other. Therefore, we developed an AiS-TSO by imitating an MCNC's united synaptic functions of reception and preprocessing of tactile information. As shown in Fig. 3a, the AiS-TSO mimics both the structure and functionality of MCNCs. The photograph of AiS-TSO and TEM image of cross-sectional view of Fe-OFET are shown in Supplementary Fig. 13, and image of experimental setting for touch measurement is shown in Supplementary Fig. 14. The gate electrode of the AiS-TSO is the receptive part to which touch were applied (Methods section). The $I_{PSC}$ response of AiS-TSO to prolonged tactile stimulation was induced by tribocapacitive potential by finger touch (Fig. 3b); the corresponding

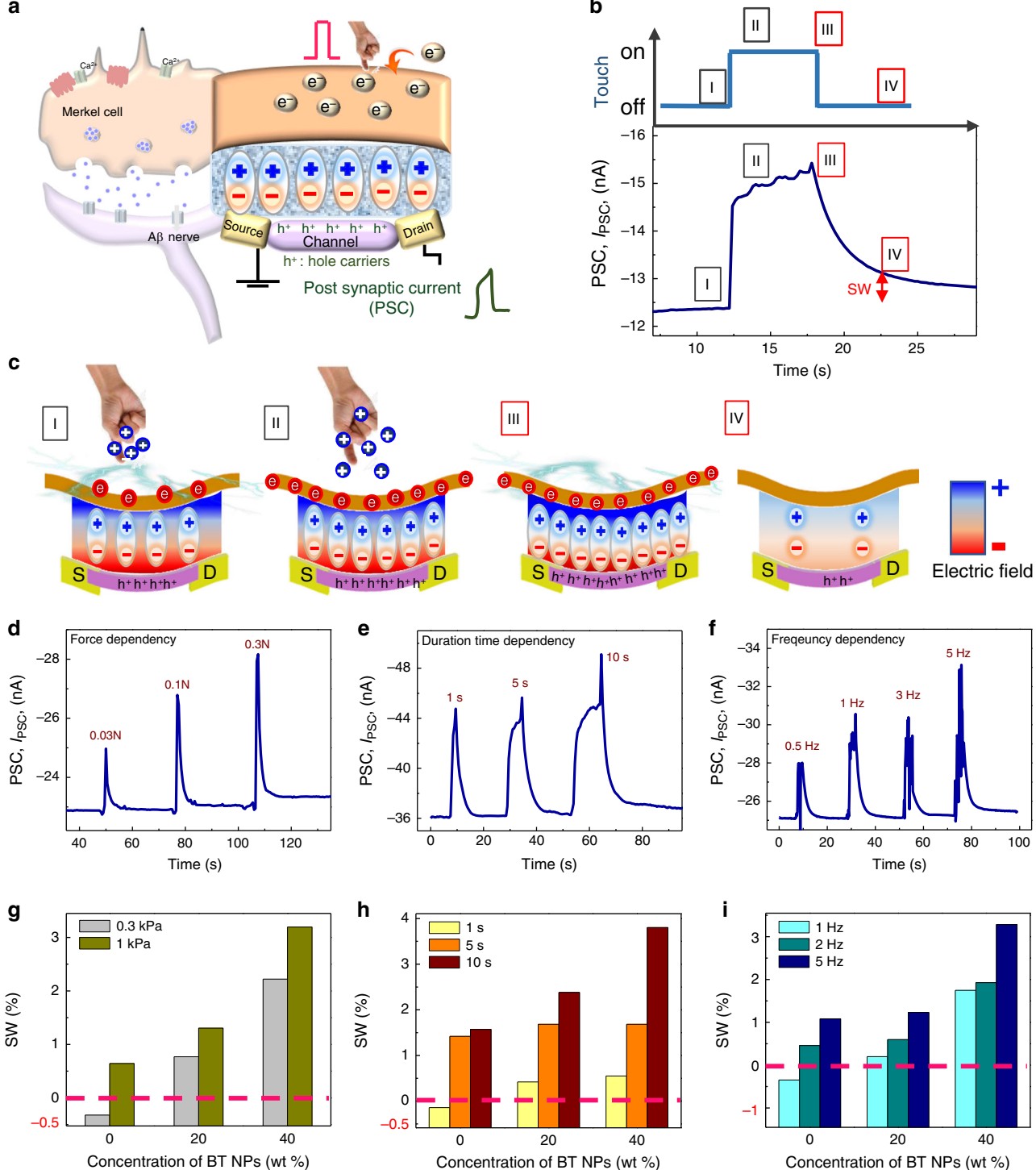

**Fig. 3 Structure, mechanism, and synaptic properties of the AiS-TSO. a** Sketch of device structure corresponding to a Merkel cell and MCNCs structure. **b** PSC during a tactile stimulus (≈0.3 kPa). A continuous bias of –3 V was applied to the drain electrode to read the PSC. **c** Schematic explanation of dipole switching in the ferroelectric layer and accumulation of hole carriers in the channel induced by tactile stimulation. Illustration of the state of I–IV of **b** and principle of formation of $P_r$ with a change in channel conductance corresponding to SW. **d** Relative changes in the PSC of the AiS-TSO at different pressures. **e** Dependence of the PSC of the AiS-TSO on touch duration (≈1 kPa). **f** Dependence of the PSC of the AiS-TSO on touch rate (≈1 kPa for 5 s). **g** Relative changes in SW for different pressures and fractions of BT NPs in the ferroelectric nanocomposite. **h** Relative changes in SW for different durations of touch and fractions of BT NPs in the ferroelectric nanocomposite (≈1 kPa). **i** Relative changes in SW for different frequencies of touch and fractions of BT NPs in the ferroelectric nanocomposite (≈1 kPa). Pink dotted line **g–i** indicates arbitrary criteria for STP/LTP and noise filtering based on SW.

working principle of AiS-TSO is shown schematically in Fig. 3c. In step I (I in Fig. 3b, c), mechanical contact of skin with the receptive part induces movement of negative charges from the skin to the receptive part because of the difference in electron affinity between them. Indeed, the current increases or decreases depending on the triboelectric properties of the material in contact with the receptive part due to different electron affinities (Supplementary Fig. 15 and Supplementary Note 8). Receptor potential of a Merkel cell accepting $Ca^{2+}$ ions forms continuously from the beginning to the end of the stimulation due to the high resistance of the membrane and is the key factor responsible for SA sensation[23]. Similar to this, in step II (II in Fig. 3b, c), the receptive part receives the pumped triboelectric charge until equilibrium is reached, resulting in tribo-capacitive potential; thus, the $I_{PSC}$ increases with a decreasing slope throughout sti-mulation, akin to SA sensation. When the tactile stimulus is removed in step III (III in Fig. 3b, c), the equilibrium is disrupted, and the negative charges on the triboelectric layer induce greater alignment of dipoles in the same direction to compensate for the removed positive charge on the skin, causing the $I_{PSC}$ to increase for a moment. The $I_{PSC}$ then decreases slowly, exhausting the tribo-capacitance stored in the receptive part. Even though the stimulus is removed, there is polarization because the ferroelec-tricity of the nanocomposite modulates the channel conductance and, in turn, generation of SW (IV in Fig. 3b, c). The SA prop-erties of the Fe-OFET were confirmed by applying a sustained $V_{rec}$ with varying amplitude for ~10 s (Supplementary Fig. 16). Here, the SW of the AiS-TSO, which we ascribe to polarization switching, is similar to neurotransmitter release at synapse-like connections in MCNCs. The SA characteristics of the AiS-TSO endow it with synapse-like connections, expanding the basic reception functions of the sensor. Also, to confirm that the main mechanism of AiS-TSO is triboelectric-capacitive coupling effect, we investigated the response of AiS-TSO to varying temperature during touch and mechanical bending strain. The results indi-cated that the pyroelectric or piezoelectric effects were negligible compared to triboelectric-capacitive coupling effect (Supple-mentary Fig. 17 and Supplementary Note 9). In addition, func-tions of AiS-TSO are clearly different from those of conventional piezoelectrically-coupled tactile sensor such as piezoelectric oxide semiconductor FET (POSFET)[53,54]. More detailed explanation is given in supplementary information (Supplementary Fig. 18 and Supplementary Note 10).

To demonstrate the synapse-like characteristics of AiS-TSO, we touched with different forces (as measured by a hand force gauge), durations, and frequencies to the device with a nanocomposite (20 wt% BT NP) layer (Fig. 3d–f, respectively). Both $I_{PSC}$ and SW during stimulation (different forces but same duration of 1 s) increased as tactile force increased, indicating a transition from STP (SW at ≈0.3 kPa force = 0.15) to LTP (SW at ≈3 kPa force = 1.86) (Fig. 3d). A larger force resulted in a larger contact area between the triboelectric layer and skin so that tribo-capacitive potential increased (Supplementary Fig. 19)[15,16,18,55–58]. These results are consistent with the electrical analysis results shown in Fig. 2. Furthermore, the effects of duration of stimulation were detected by monitoring both $I_{PSC}$ and SW. When touch with the same force (≈1 kPa) but different durations (1 s/5 s/ 10 s) were applied, the SW for a stimulus duration of 10 s (SW = 2.4) was five times larger than that for a duration of 1 s (SW = 0.42) (Fig. 3e). The longer the stimulus was applied, the greater was the triboelectric charge pumped from the skin to the receptive part, which resulted in greater alignment of ferroelectric dipoles and hence a larger SW. In biological synapses, SRDP is related to the boosted signal from consecutive pulses with a narrow interval by residual neurotransmitters released as a result of the previous pulse[59]. We expect that MCNCs have a similar characteristic as

this would help distinguish significant and repetitive stimuli from noise. Therefore, we created an AiS-TSO filtering function by setting a SW threshold and performing measurements (Fig. 3f). When we touched the device at a frequency of 0.5, 1, 3, or 5 Hz for a duration of about 5 s under manual forcing at ≈1 kPa, SW increased. If the threshold filtering frequency is set using SW value, the AiS-TSO can be utilized as an intelligent sensor at the unit device level. We further investigated the characteristics of AiS-TSO such as PSC change ratio ($\Delta I_{PSC}/I_{PSC,i}$) by repetitive touch with different retention time and PPR with different time interval of touch (Supplementary Fig. 20). We observed that the shorter retention time made the $\Delta I_{PSC}/I_{PSC,i}$ larger. Also, PPR was increased when the time interval of two consecutive touches was decreased. By the way, since the triboelectric effect is highly influenced by humidity[60,61], we investigated the PSC response of AiS-Tso by finger touch in environments with different humidity levels. The results indicated that the $I_{PSC}$ value was decreased with the humidity level increased (Supplementary Fig. 21).

In addition, we were able to tune the filtering functions of AiS-TSO by varying the fraction of BT NPs in the nanocompo-site. BT NPs in the various fraction ferroelectric nanocomposites were well dispersed (Supplementary Fig. 22). When we applied 0.3 kPa (duration of 1 s), the SW of the device with P(VDF-TrFE) only was −0.32. However, the SW of the nanocomposite (40 wt% BT NP) device was 2.22 (Fig. 3g). Therefore, if the threshold of noise filtering is set to be a positive SW value, the signal at the applied force of 0.3 kPa is filtered out as noise for the device with P(VDF-TrFE) only. However, in the nanocom-posite (40 wt% BT NP) device, both signals from the device at forces of ≈0.3 and ≈1 kPa (SW = 3.19) were treated as mean-ingful. To investigate the dependency of the filtering function on fraction of BT NPs, we touched the device with a pressure around 1 kPa for 1, 5, or 10 s. As shown in Fig. 3h, SW increased as the weight fraction of BT NPs increased from 0 to 20 to 40 wt % (SW values of −0.14, 0.42, and 0.55, respectively) for a stimulus duration of 1 s. When we set the criterion of STP to SW = 0, the device with P(VDF-TrFE) only showed STP for a duration of stimulation of 1 s. For all three cases, SW values with 10 s stimulation were 1.6, 2.4, and 3.8 for 0 wt%, 20 wt%, and 40 wt% BT NPs, respectively, which were 5 times higher than those obtained for a 1-s stimulation. When we set the criterion for LTP to SW = 3, only the AiS-TSO with BT NPs (40 wt%)/P(VDF-TrFE) showed LTP characteristics. Dependence of filtering function on the frequency of touches for devices with different fractions of BT NPs was observed (Fig. 3i). When we set the criteria of noise filtering to a positive SW, no signals of the device with a BT NP(40 wt%)/P(VDF-TrFE) layer were filtered (SW = 1.8, 1.9, and 3.3 for 1, 2, and 5 Hz, respectively). For the device with P(VDF-TrFE) only, however, signals from 1 Hz (SW = −0.34) were treated as noise. These data indicate that a wide range of filtering parameters can be developed based on the tunability of the SW of AiS-TSO. Using different SW change depending on the concentration of BT NPs in the nanocompo-sites, therefore, we could set the different criteria to be used for noise filtering or getting specific range of information similarly to biological mechanoreceptors which transfer signals to brain depending on characteristics of the receptor cells or number and distribution of connections between afferent neurons[12].

In human sensory perception systems, synapses play an important role in memory functions. As seen from the schematic in Fig. 4a, the learning and memory model proposed by Atkinson and Shiffrin classifies memory into three steps: sensory memory (SM), short-term memory (STM), and LTM[62]. SM is a memory temporarily stored in the human brain about incoming sensory information that is initially processed and then transferred to short-term storage. STM is related to the STP of the SW and is

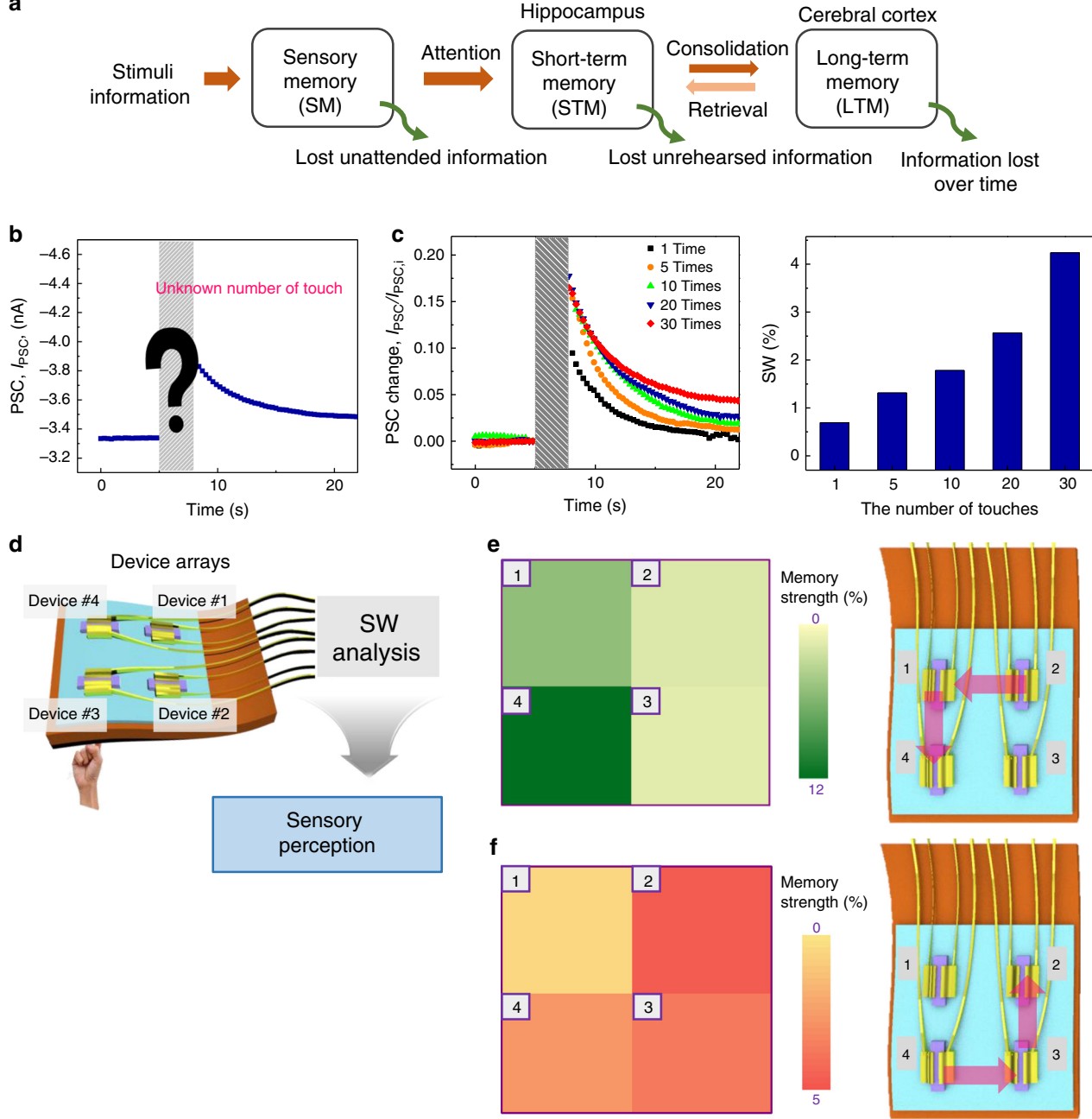

**Fig. 4 Intelligence of an AiS-TSO with sensory memory. a** A typical model of learning and memory processing in human neural systems illustrating sensory-memory, short-term memory, and long-term memory behaviors. **b** The PSC of the AiS-TSO in response to tactile stimulation with an unknown number of stimuli. **c** Relative PSC change ratio ($\Delta I_{PSC}/I_{PSC,i}$) according to number of touches (frequency, 1 Hz and pressure, ≈1 kPa of touch) (left). The real-time PSC during the stimulation was obscured. SW of the AiS-TSO according to number of touches right). **d** Schematic illustrations of the AiS-TSO showing reception and preprocessing in parallel and the intelligence, including sensory memory, afforded by analysis of SW. **e, f** Schematic illustration of mapping of the memory magnitude of pixels in a 2 × 2 array of AiS-TSOs and the expected order (2 → 1 → 4→ for **e**, 4 → 3 → 2 for **f**) of touched pixels based on memory magnitude, which is related to SW.

responsible for moment-to-moment awareness of environment changes. If the stimulation is repeated, then it becomes an LTM in the cerebral cortex, a process referred to as consolidation[62]. Many researchers have investigated the functions of STM or LTM using synaptic devices based on brain synapses[32,63–67], but there is a lack of research into sensory perception based on sensory neural synapses. Here, we mimicked this memory process using synapse-like connections in our sensor, which is a very promising first-step to realizing memory functions in bio-inspired artificial

sensory systems, which would mitigate the requirement for further information processing.

Based on the structure of the sensory organs, we implemented a memory function in an AiS-TSO without any other storage device based on synaptic plasticity. Even without knowledge of the real-time signal of $I_{PSC}$ during touching, the number of touches of similar force can be inferred by analyzing signals after touching (Fig. 4b). $I_{PSC}$ after a touch increased as the number of touches increased from 1 to 30 (≈1 kPa and frequency of 1 Hz)

(left, Fig. 4c). Here, SW values ranged from 0.6 to 4.2 when the number of touches increased from 1 to 30 (right, Fig. 4c). Therefore, SNDP in AiS-TSO observed with finger touch (Fig. 4c and Supplementary Fig. 23) shows potential to be used as a sensing device embedded with memory function at unit device level without the need for an additional memory device. To emulate the memory process of a sensory organ, tracking the order of touch is also important. We constructed a $2 \times 2$ array of AiS-TSO pixels (Fig. 4d) and touched three pixels in sequence with the same force for 5 s with an interval of 15 s between the different pixels. We then analyzed the degree of memory extracted from the $I_{PSC}$ before ($I_{PSC,0}$) and after touch ($I_{PSC,mem}$), described as $(I_{PSC,0} - I_{PSC,mem})/I_{PSC,0}$. Because the pixels were touched sequentially, not simultaneously, there were differences in the time interval between $I_{PSC,0}$ and $I_{PSC,mem}$ depending on the order of touch (~77, ~67, and ~57 s for the 1st, 2nd, and 3rd pixels touched). The memory strength for the non-touched pixel was smallest, and that of the last touched pixel was the largest because the decrease in $I_{PSC}$ was shortest for the last pixel. Based on an analysis of the strength of memory, we derived the order of touch from the stored tactile information in SW. As shown in Fig. 4e, the strength of memory of each pixel from 1 to 4 was 4.7, 1.6, 1.3, and 11, respectively. Therefore, we postulated that pixels were touched in the order of pixel $2 \rightarrow 1 \rightarrow 4$, and that pixel 3 was not touched. The results shown in Fig. 4f indicate that the touch order of the pixels was in fact $4 \rightarrow 3 \rightarrow 2$, and that pixel 1 was not touched. This result corresponds to the strength of memory of pixels 1, 2, 3, and 4, which was 1, 5.3, 4.2, and 3.1, respectively. Also, we measured the memory strength three times with the touch order of pixel $2 \rightarrow 1 \rightarrow 4$, without touching of pixel 3, showing the same tendency for all three measurements (Supplementary Fig. 24).

## Discussions

Our AiS-TSO successfully imitates the intelligent functions of a human touch sensory organ, namely the MCNC, due to a simple synapse-like connection at the unit sensor level. We utilized a tribo-capacitive coupling effect in a ferroelectric transistor to endow the sensor with synaptic functions. We were able to control the synaptic characteristics of the AiS-TSO by tuning SW through variation of the BT NP fraction in the ferroelectric nanocomposite gate dielectric layer. This study shows a simple demonstration of filtering function by fabricating Fe-OFETs with the ferroelectric nanocomposite with the same concentration of BT NPs on a single substrate. AiS-TSO would be more practical if it is possible to fabricate an array of devices patterned with nanocomposites of different of BT NP concentrations by using an additive printing process of improved nanocomposite solutions. Since AiS-TSO is flexible, it can be applied to soft biomimetic devices in applications such as intelligent soft electronic skins. We demonstrated that our AiS-TSO device has adaptation, filtering, and memory functions and shows parallel spatiotemporal reception and preprocessing of tactile information by SW control using the partial polarization of minor loops in ferroelectric layer. We can also consider saturation loop of ferroelectric layer for future work to get synaptic properties with longer retention time. We successfully mimicked the synapse-like connection of receptor cells and afferent neuron terminals in a sensory organ, thereby endowing the sensor itself with intrinsic intelligence without linking it to a neuronal processor. We believe that this flexible AiS-TSO inspired by the MCNC represents a new paradigm in sensor technology for neuro-robotics, autonomous systems, intelligent electronic skins and better AI technologies that decreases the requirement for information processing and facilitates greater device integration and energy efficiency.

## Methods

**Preparation of materials and device fabrication.** Materials used in the ferroelectric nanocomposites were P(VDF-TrFE) (65 mol% VDF) purchased from Piezotech S.A. and BT NPs purchased from Sigma Aldrich. The BT NPs were ball-milled for 30 min to enhance their dispersion in solution. Then, the BT NPs were dissolved in a 3-aminopropyltriethoxy silane (APTES)/ethanol solution. The solution was adjusted to a pH of 4–5 using acetic acid and stirred for 12 h. The solution was then centrifuged, and the BT NPs were washed with ethanol to remove residual APTES. The filtered BT NPs were cured at 110 °C for 10 min in a small amount of ethanol and then mixed with $N,N$-dimethylformamide (DMF, purchased from Sigma Aldrich). After mixing, the solution was centrifuged again to obtain a solvent-particle mixture with BT NPs. BT NPs at a specified fraction (wt %) with respect to P(VDF-TrFE) were dispersed in DMF with P(VDF-TrFE). To fabricate Fe-OFET, a 100-nm-thick Ni gate electrode was deposited on a clean PI substrate (75 µm-thick) by electron beam evaporation. The prepared ferroelectric nanocomposite solution was spin-coated as the gate dielectric layer onto the Ni gate electrode, followed by drying at 110 °C to remove the DMF solvent. Next, the gate dielectric layer was annealed at 200 °C to melt it completely. Annealing was continued in nitrogen ambient at 140 °C for β-phase crystallization of P(VDF-TrFE), which contributes to polarization. We did not conduct poling process for gate insulator. Pentacene as a channel layer (70 nm) and Au (70 nm) as the S/D of the Fe-OFET were deposited by thermal evaporation. The process of AiS-TSO fabrication was the same except that the Fe-OFET and the gate electrode were not formed and a thinner PI substrate (50 µm-thick) was used to enhance the triboelectric effect.

**Measurements.** Electrical characteristics and the PSC in response to applied electrical pulses and finger touch stimuli were measured using a semiconductor parameter analyzer (Keysight, B1500). Touch stimuli force was measured using a hand force gauge (Algol instrument, HF-10), and the output voltage of triboelectrification with finger and PI was measured with an oscilloscope (Tektronix, TDS 3014B). Dynamic and static flexibility tests of devices were carried out using a custom-built bending system. Dispersion of loaded BT NPs in the ferroelectric nanocomposite and the thickness of nanocomposite were examined by field-emission scanning electron microscopy (FE-SEM, JEO JSM-6500F). Capacitance measurement with MIM device was conducted using a semiconductor parameter analyzer (Agilent B1500).

## Data availability

The data that support the findings of this study are available from the corresponding author upon request.

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

## Acknowledgements

This work was supported by the Basic Science Research Program (2020R1A2C3013480 & 2019R1A6A1A03033215) through the National Research Foundation (NRF). The work was also partially supported by Samsung Electronics. We would like to thank J.-Y. Kim, Y.-I. Choi, and M.-J. Choi at SKKU and Dr. S. Siddiqui at BUITEMS for their help with material preparation and characterization.

## Author contributions

Y.R.L. and N.-E.L. designed the experiments. Y.R.L. conducted experiments, fabricated devices, and analyzed data. Y.R.L. and T.Q.T. measured the electrical properties of devices. Y.R.L and B.-U.H. conducted flexibility tests of the devices. All authors contributed to the manuscript by discussion of the results. Y.R.L. and N.-E.L. wrote this article.

## Competing interests

The authors declare no competing interests.
