## [Peer Review File · Nature Communications]

Reviewers' comments:

Reviewer #1 (Remarks to the Author):

When encoding external stimuli for Merkel neural cluster, not only the curvature of the external force application object but also the depth of press-in, but also the influence at different positions from the point of stimuli. Therefore, for Merkel neural cluster, two main problems are studied: first, when external stimulation is unchanged, the release mode of the sensor neurons in different positions; The second is the overall distribution of the whole Merkel cell cluster when external load forces of different intensities are applied.

This paper is of great interest in the sense of engineering, but the theoretical support is not sufficient. It is suggested that the following articles should give sufficient attention to the authors so that the physical application of the Merkel cell has a clear biological basis.

(1) Mengqiu Yao, Rubin Wang, Neurodynamic analysis of Merkel cell–neurite complex transduction mechanism during tactile sensing. *Cognitive Neurodynamics*. June 2019, Volume 13, Issue 3, pp 293–302

(2) Hu Jiyong, Zhao Qun, Jiang Ruitao, Wang Rubin, Ding Xin. Responses of cutaneous mechanoreceptors within fingerpad to stimulus information for tactile softness sensation of materials. *Cognitive Neurodynamics*, 2013, 7(5): 441-447,

(3) Hu Jiyong, Zhang Xiaofeng, Yang Xudong, Jiang Ruitao, Ding Xin Wang Rubin, Analysis of fingertip/fabric friction-induced vibration signals toward vibrotactile rendering, *The Journal of The Textile Institute*, 2016, 107(8): 967-975,

Reviewer #2 (Remarks to the Author):

This manuscript titled "A Flexible Artificial Intrinsic-Intelligent Tactile Sensory Organ" by Y. R. Lee et al. deals with a flexible tactile sensory-organ mimicking the synapse-connections of Merkel cells with neurons with intrinsic intelligence of the sensor that latter being based on a flexible ferroelectric OFET gated via a touch-induced triboelectric-capacitive coupling.

The topic is very interesting for the field of neuro-robotics and epidermal electronics. The main novelty lies in the claim of a triboelectric-capacitive coupling effect to the ferroelectric transistor (AiI-TSO = Fe-OFET without a gate electrode) upon human touch endowing synaptic functions like reception, filtering and memory.

However, it is not clear whether the observed effects in the transistor are really due to triboelectric charge transfer to the receptive part (polyimide film) or to the standard piezoelectric and pyroelectric effects that are intrinsic to ferroelectric materials, especially as the authors claim that the device is "naturally" poled upon processing. In other words: How can you differentiate the claimed triboelectric effect from the piezo- and or pyroelectric effects seen in ferroelectric polymer sensors or Fe-OFETs? In the latter case any touch or pressure applied to the ferroelectric device will result in a change of the dipole-density in the ferroelectric layer and thus a generation of compensation charges at the uppermost interface and a drain current, which decays exponentially and has a very similar signal form as is observed e.g. in Fig. 2f, Fig. 3d,e... Even the increase/decrease of the I_{psc} depending on the contacting material (e.g. skin or PEN) can also be easily be attributed to pyroelectricity when the different thermal conductivities of the individual contacting materials are taken into account.

If this concern can be answered satisfactory and the difference to piezoelectricity proven, I recommend publishing the manuscript after major revision also covering the issues listed below:

1. Please specify all acronyms in the figure captions
2. Please specify the organic semiconductor in the main text of the manuscript.
3. Please indicate the thickness of the PVDF-TrFE/BC nanocomposite gate dielectrics.
4. Please indicate the magnitude of the coercive fields for Fe-OFETs with and without BT NPs, it is only stated that the composites have a smaller coercive field than pure PVDF-TrFE.
5. A permanent polarization (which is claimed to be responsible for SW with LTP behavior due to a longer retention time of permanent polarization) can only occur, if the touch-induced

voltage/potential is larger than the coercive field of the ferroelectric dielectric. In order to assess if this is really the case, it would be decisive to know the thickness of the pure PVDF-TrFE layer at least, which typically has a coercive field of $\sim 50\text{MV/m}$. Please provide more information on this. Furthermore, permanent polarization means that the polarization is permanent or, as it is mostly called, remanent and does not show any retention time.

6. Could it be that the LTP behaviour, the increase of residual current or decay time with frequency (or number of pulses over a specific time period) is rather due to charge trapping in the OSC layer than to change in the "permanent" polarization?

7. It is unclear, how the intrinsic hysteresis effects in the Fe-OFETs, especially for the composites, will influence the PSC signals and their reproducibility (see SFig.1). In Fig.2f only the influence of increasing the Vrec amplitude (related to a claimed SADP) was investigated, but please show if this SADP effect is quantitatively reproducible also for decreasing the Vrec amplitude from -20V to -1V (especially for the composite material).

8. Is the SNDP effect the same, if the number of spikes are decreased from e.g. 100 to 1 (Fig. 2g) ? This is important in order to judge the reproducibility and usefulness of the signals.

9. What about gate leakage currents in the Fe-OFETs?

10. It has to be clearly stated that behaviour of the AiI-TSO under tensile strain was tested only for strains up to 1.88% which is far from what is the tensile strain strength of the skin in a human body ($\sim 25\%$, e.g. A. Gallagher, IRC12-59, IRCOBI Conference). So the analogy of the proposed TSO with MCNCs is very limited in this regard. By the way, under repetitive compressive strain SFig 6e there is a large difference between initial values of I_{psc} and compressed values.

11. Please explain, how the claimed tunable filtering ability in terms of SW can be implemented in a real-world artificial TSO, since it is mainly based on using ferroelectric materials with different NP concentration.

12. In Fig.4e it is not clear why it can be deduced that the memory strength of Pixel 3 of 1.3 means clearly that the Pixel 3 was not touched, whereas a strength of 1.6 for Pixel 2 means that the Pixel 2 is touched. The difference seems to be too small to make an unambiguous decision.

13. Please remove the term "self-powered" since the device clearly needs the application of a potential to the drain electrode, in order to be functional.

14. The influence of repetitive pulses or number of pulses on the I_{psc} (SNDP) or paired pulses was not investigated for the AiI-TSO. What is expected here?

Reviewer #3 (Remarks to the Author):

This paper reports a sensor that utilizes a ferroelectric composite as dielectric transistor material.

In general, the wording used in this manuscript is often times complicated, difficult to read, due to punctuation errors, and misleading.

Some examples are:

Title: "A Flexible Artificial Intrinsic-Intelligent Tactile Sensory Organ ", which claims an "intelligent" sensory organ, whereby it is completely unclear how the demonstrated sensor is "intelligent", i.e. has the ability to acquire and apply knowledge and skills (from Oxford Dictionary);

or

"Sensory organs enable animals to gather information to conduct skilled movements...", where sensors are linked to skilled movements, while one has nothing to do with the other;

or

"Even though much effort has been made to mimic mechanoreceptors in the human body, emulation of their intelligent functions to extend sensory reception in an efficient way has not been widely successful. ", which is a generic statement that lacks explanation what "in an efficient way" means and where "... has not been widely successful" is also a completely vague statement.

The meaning of sentences like "Touch stimuli control alignment of permanent dipoles in the ferroelectric gate dielectric so that the output signal is endowed with tactile information with the

parallel functions of slowly adapting (SA) sensation, filtering, and memory in a self-powered manner. " is obviously difficult to understand, due to missing words etc.

Characterization of the ferroelectric composite is absent. The minimum should be a polarization curve (P over E) to see the remanent polarization and permittivity of the material. Has the composite been exposed to any poling? Otherwise, the dipole alignment would be random. Hysteresis measurements of the polarization for the applied electric field would be very interesting too, in order to understand the hysteresis effect on the gate potential.

The described concept has previously been called POSFET touch sensitive devices that use piezoelectric polymers for tactile sensing. Therefore, the novelty of this work is not obvious. Especially the title implies a new concept, which, does not seem to be justified.

The authors show very well the ability to adjust the synaptic weight by varying the filler content. However, using this method the weight is defined during the fabrication and no changes can be made afterwards, compared to the real synapses that have the ability to strengthen or weaken the weight. This synaptic plasticity is a crucial ability for learning and adaptation. On the other hand, it seems that the synaptic weight could also be adjusted by adjusting the gate voltage, as shown in this paper, a parameter that is much easier to control in an artificial system, and which could be adjusted over time in a learning-like manner. An "imprinted" synaptic weight could also be achieved during fabrication by other means like channel width.

An interesting issue of the proposed approach is the practicality when considering array fabrication, where I imagine it very difficult to adjust the filler concentration from one transistor to another on the same wafer, in order to obtain the desired weights.

The manuscript shows many schematic images of the device, but not one real photograph, SEM, TEM etc. image.

It would be interesting to see the effect of moisture on the pressing finger. Is there any effect of the high permittivity of water on the function of the tactile sensor?

It seems that the response of the sensor to constant pressure is variable. How could this be fixed?

The force measurement seems very basic using a hand force gauge. How did the authors ensure a constant force over time or a repetitive force of equal amplitude in case of varying forces?

RESPONSE TO REFEREES (NCOMMS-19-17002)

Reviewer #1

1. When encoding external stimuli for Merkel neural cluster, not only the curvature of the external force application object but also the depth of press-in, but also the influence at different positions from the point of stimuli. Therefore, for Merkel neural cluster, two main problems are studied: first, when external stimulation is unchanged, the release mode of the sensor neurons in different positions; The second is the overall distribution of the whole Merkel cell cluster when external load forces of different intensities are applied.

This paper is of great interest in the sense of engineering, but the theoretical support is not sufficient. It is suggested that the following articles should give sufficient attention to the authors so that the physical application of the Merkel cell has a clear biological basis.

(1) Mengqiu Yao, Rubin Wang, *Neurodynamic analysis of Merkel cell–neurite complex transduction mechanism during tactile sensing. Cognitive Neurodynamics. June 2019, Volume 13, Issue 3, pp 293–302,*

(2) Hu Jiyong, Zhao Qun, Jiang Ruitao, Wang Rubin, Ding Xin. *Responses of cutaneous mechanoreceptors within fingerpad to stimulus information for tactile softness sensation of materials. Cognitive Neurodynamics, 2013, 7(5): 441-447,*

(3) Hu Jiyong, Zhang Xiaofeng, Yang Xudong, Jiang Ruitao, Ding Xin Wang Rubin, *Analysis of fingertip/fabric friction-induced vibration signals toward vibrotactile rendering, The Journal of The Textile Institute, 2016, 107(8): 967-975,*

Response:

The comments on insufficient theoretical information about Merkel neural clusters are greatly appreciated.

As commented by the reviewer, important were added as references, 12,13 and 45.

12. Hu, J., Zhao, Q., Jiang, R., Wang, R. & Ding, X. Responses of cutaneous mechanoreceptors within fingerpad to stimulus information for tactile softness sensation of materials. *Cogn. Neurodyn.* **7**, 441–447 (2013).
13. Hu, J., Zhang, X., Yang, X., Jiang, R. Ding, X. & Wang, R. Analysis of fingertip / fabric friction-induced vibration signals toward vibrotactile rendering. *J. Text. Inst.* **107** 967-975 (2016).
45. Yao, M. & Wang, R. Neurodynamic analysis of Merkel cell–neurite complex transduction mechanism during tactile sensing. *Cogn. Neurodyn.* **13**, 293–302 (2019).

The further explanation on relations between MCNCs and SA firing which could be imitated as SA sensors (including refs. 12 and 13) were added in page 2 as follows: “There have been some reports on artificial tactile sensors mimicking mechanoreceptors with slowly adapting (SA) firing analysis^{12,13}.”

Also, to support the necessity of mimicking of filtering functions with investigation of MCNCs and SA firing, we added the sentence in page 11 as follows: “Using different SW change ratio depending on the concentration of BT NPs in the nanocomposites, therefore, we could set the different criteria to be used as noise filtering or getting specific range of information similarly to mechanoreceptors which transfer signals to brain depending on characteristics of the receptor cells or number and distribution of connections between afferent neurons¹².”

We also corrected the sentence describing the research on MCNCs and SA perceptions in page 8 as follows : “There is still active research about structural and phenomenological observations on MCNCs^{11,23,24,44} including the relationship between the number and spatial density of MCNCs and SA perception⁴⁵. Although the exact mechanism of SA firing in MCNCs has not been discovered, it is obvious that there are complex interactions between mechanosensitive Piezo-2 channels, cell membrane potentials, and synergistic synapses of MCNCs that allow Merkel cells to initiate A β afferent pulses to encode tactile information^{9,22,46},”

Reviewer #2:

This manuscript titled “A Flexible Artificial Intrinsic-Intelligent Tactile Sensory Organ” by Y. R. Lee et al. deals with a flexible tactile sensory-organ mimicking the synapse-connections of Merkel cells with neurons with intrinsic intelligence of the sensor that latter being based on a flexible ferroelectric OFET gated via a touch-induced triboelectric-capacitive coupling.

The topic is very interesting for the field of neuro-robotics and epidermal electronics. The main novelty lies in the claim of a triboelectric-capacitive coupling effect to the ferroelectric transistor (AiI-TSO = Fe-OFET without a gate electrode) upon human touch endowing synaptic functions like reception, filtering and memory.

However, it is not clear whether the observed effects in the transistor are really due to triboelectric charge transfer to the receptive part (polyimide film) or to the standard piezoelectric and pyroelectric effects that are intrinsic to ferroelectric materials, especially as the authors claim that the device is “naturally” poled upon processing. In other words: How can you differentiate the claimed triboelectric effect from the piezo- and or pyroelectric effects seen in ferroelectric polymer sensors or Fe-OFETs? In the latter case any touch or pressure applied to the ferroelectric device will result in a change of the dipole-density in the ferroelectric layer and thus a generation of compensation charges at the uppermost interface and a drain current, which decays exponentially and has a very similar signal form as is observed e.g. in Fig. 2f, Fig. 3d,e... Even the increase/decrease of the I_{psc} depending on the contacting material (e.g. skin or PEN) can also be easily be attributed to pyroelectricity when the different thermal conductivities of the individual contacting materials are taken into account.

Response

We appreciate the comments on underlying the possible mechanism of AiS-TSO is pyro- or piezo- or tribo-electric capacitive coupling effect. To demonstrate the main mechanism of AiS-TSO is triboelectric-capacitive effect, we carried out additional experiments on touching of AiS-TSO with the nanocomposite (BT NPs 20 wt%) by finger or polyimide (PI) which is the same material as the sensing part while varying

the temperature (to check the effect of pyroelectricity) and applied bending strain (to check the effect of piezoelectricity). When we investigated the effect of elevated temperature on responses of the AiS-TSO to finger touch, we found that I_{PSC} was increased as the temperature is increased from 27 to 35 °C. When the AiS-TSO was touched by PI with the temperature varied from 25 to 30 °C, there was only a slight increase in PSC (**Supplementary Fig. 15a**), which indicate a small thermal response of AiS-TSO. However, much larger change ratio of PSC with finger touch indicates a triboelectric-capacitive coupling as a main mechanism. Change in the synaptic weight by PI touching was also negligible compared to that of finger touch.

To figure out the effect of thermal stimuli in more detail, AiS-TSO with the nanocomposite (BT NPs 20 wt%) and OFET with PVP (polyvinylpyrrolidone) gate dielectric with no ferroelectricity and pyroelectricity were also touched by finger with the temperature of finger varied (27 – 35 °C). As seen from the results (**Supplementary Fig. 15b**), the OFET with PVP showed a small change in the drain current presumably due to an increase in channel conductance induced by thermally activated carriers in semiconductor channel while AiS-TSO with the nanocomposite (BT NPs 20 wt%) showed much larger change in PSC than that of the OFET with PVP. And, the OFET with PVP responded to thermal stimuli with increasing PSC (drain current), which indicates that temperature change in addition to touch increased the conductance increase in semiconductor channel at the elevated temperature presumably due to thermal generation of carriers (**Supplementary Information refs.14 and 27**). From the results from OFET with PVP gate dielectric, it can be concluded that the observed response of AiS-TSO to touch + thermal stimuli is not attributed to pyroelectricity of the nanocomposite gate dielectrics and much smaller than due to triboelectric-capacitive coupling effect.

In order to investigate the effect of piezoelectricity of ferroelectric nanocomposite, AiS-TSO was tested under tensile and compressive bending. As seen from the data of PSC change ratio (**Supplementary Fig. 15c**), the PSC change ratio by finger touch was much larger than that by compressive and tensile bending. Increased in PSC change upon compressive strain may be attributed to piezoresistive property of pentacene channel where carrier mobility is increased under compressive strain and vice versa under tensile strain (**Supplementary Information ref. 27**). As a result, there was no observed the synaptic weight (SW) change by

piezoelectric effect of the ferroelectric nanocomposite in AiS-TSO, which indicates that SW change by finger touch was mainly occurred due to the triboelectric-capacitive coupling effect in the device.

Supplementary Figure 15. Responses of AiS-TSO to thermal and strain stimuli. a, change ratio of PSC ($\Delta I_{PSC} / I_{PSC,i}$) of AiS-TSO when touched by polyimide (PI) film and finger with varying temperature of polyimide and finger. The devices were touched with stimulation time of ≈ 3 s at the force of ≈ 1 kPa. **b**, PSC (I_{PSC}) of AiS-TSO with BT NP(20 wt%)/P(VDF-TrFE) gate dielectric (blue) and OFET with PVP gate dielectric (red) with the temperature of finger varied during finger touch. The devices were touched by finger with stimulation time of ≈ 3 s at the force of ≈ 1 kPa. **c**, change ratio of PSC ($\Delta I_{PSC} / I_{PSC,i}$) in AiS-TSO under touching (stimulation time of ≈ 10 s at the force of ≈ 1 kPa) and bending strains (bending radius of 2 mm).

For further investigation about the mechanism of our AiS-TSO, we characterized the response of PSC depending on touching object, temperature and bending strain. First, we touched the AiS-TSO with

polyimide (PI) film and finger varying the temperature. As shown in **Supplementary Figure 15a**, the PSC change ($\Delta I_{\text{PSC}}/ I_{\text{PSC},i}$) of AiS-TSO to finger touch was much larger by three orders of magnitude than that to touching with PI at the same temperature ($\sim 27^\circ\text{C}$). In both cases, the $\Delta I_{\text{PSC}}/ I_{\text{PSC},i}$ was increased with the temperature increased. These results indicate that the main mechanism of AiS-TSO is triboelectric-capacitive coupling effect even though there is a slight change in the $\Delta I_{\text{PSC}}/ I_{\text{PSC},i}$ with the temperature increased.

For further investigation effect of temperature change on the response of PSC, we fabricated the OFET device using PVP (polyvinylpyrrolidone) as gate dielectric layer which has no pyroelectricity compared its response to the AiS-TSO. As shown in **Supplementary Figure 15b**, when we touch the both devices by finger using BT NP(20wt%)/P(VDF-TrFE) and PVP gate dielectrics, we could observe the increase of PSC (I_{PSC}) in both devices with the temperature increased. From those results, it can be confirmed that the response of AiS-TSO is not originated from pyroelectricity. Since the OFET with PVP gate dielectric shows an increase in the I_{PSC} with the temperature increased, an increase in I_{PSC} may be attributed to increase in channel conductance due to thermal generation of carriers^{14,27}. Furthermore, pyroelectric effect is expected to be negligible because we didn't carry out any polling process for generating pyroelectricity. From those data, we could confirm that the pyroelectric effect in AiS-TSO is negligible compared to triboelectric-capacitive effect which is the main mechanism of touch response.

In order to investigate the response of AiS-TSO with BT NP(20 wt%)/P(VDF-TrFE) gate dielectric to mechanical strain, we also measured $\Delta I_{\text{PSC}}/ I_{\text{PSC},i}$ of the device to tensile and compressive bending strains and compared to that to finger touch. As shown in **Supplementary Figure 15c**, when the AiS-TSO is touched by finger and bent with the bending radius of 2 mm for around 10 s, The $\Delta I_{\text{PSC}}/ I_{\text{PSC},i}$ under finger touch was much larger compared to that under bending. Since the bending strain does not induce triboelectric-capacitive effect, only a small response was observed with no synaptic weight. Observed increase and decrease in $\Delta I_{\text{PSC}}/ I_{\text{PSC},i}$ of the AiS-TSO may be attributed to piezoresistive effect in the pentacene channel due to increase and decrease of hole carriers in the semiconductor channel under compressive and tensile bending strain, respectively²⁷. With a small change in PSC under bending, the device could not generate synaptic weight. These results also indicate that main mechanism of synaptic

weight generation is triboelectric-capacitive coupling effect.

Also, to indicate this result and discussion, we added the manuscript sentences in page 9 as follows:

“Also, to confirm that the main mechanism of AiS-TSO is triboelectric-capacitive coupling effect, we investigated the response of AiS-TSO to varying temperature during touch and mechanical bending strain. The results indicated that the pyroelectric or piezoelectric effects were negligible compared to triboelectric-capacitive coupling effect (**Supplementary Figure 15**).”

If this concern can be answered satisfactory and the difference to piezoelectricity proven, I recommend publishing the manuscript after major revision also covering the issues listed below:

1. Please specify all acronyms in the figure captions.

Response:

We appreciate the comment on figure caption. Short-term and long-term plasticity and post synaptic current are mentioned for the first time in captions and so we corrected the captions in **Fig. 2** to specify the acronyms as follows: “**Fig. 2. Structure, mechanism, and post-synaptic current (PSC) of the Fe-OFET to analysis of synaptic properties such as short term (STP), long term plasticity (LTP) and paired pulse ratio (PPR).** **a**, Sketch of device structure. Pre-synaptic pulse corresponds to an electrical gate bias voltage, V_{rec} . Gate dielectric layer consists of BT NPs/P(VDF-TrFE) ferroelectric material. Fraction of BT NPs can be modified to tune the SW of the PSC. **b**, Mechanism by which the PSC is generated. Schematic explaining the alignment of ferroelectric dipoles before applying V_{rec} (left), during application of a negative V_{rec} (middle), and after removal of the V_{rec} (right). **c**, Schematic explanation of SW and PPR. **d, e** STP (frequency of 0.1 Hz, **d**) and LTP (frequency of 1.42 Hz, **e**) were triggered by applying -10 V V_{rec} pulses. PSC dependence on **f**, V_{rec} amplitude (pulse width, 500 ms). **g**, number of V_{rec} pulses (amplitude of -10 V and pulse width of 500 ms). **h**, PSC before (black) and after (red) a cyclic bending test. PSC in response to a pulse and successive pulses of -10 V V_{rec} with a pulse width of 500 ms (pulse interval of successive identical pulses, $\Delta t_{rec}= 500$ ms). **i**, Variations in SW change ratio and PPR under different static tensile strain. PSC of

devices with BT NP(20 wt%)/P(VDF-TrFE) (red) and P(VDF-TrFE) only (blue) **j**, as a function of V_{rec} pulse duration. (amplitude of V_{rec} , -10 V) **k**, at different frequencies of V_{rec} (amplitude of -10 V).”

Also, the terms have been unified to make sure that the readers understand them better. We edited the terms of all “tactile stimuli” to “**touch**” in manuscript.

2. *Please specify the organic semiconductor in the main text of the manuscript.*

Response:

We specified the organic semiconductor in the main text of the manuscript in page 4 as follows: “We first fabricated and characterized an Fe-OFET device with BT NP(20 wt%)/P(VDF-TrFE) nanocomposite gate dielectric (thickness of 0.6 μm) and Ni gate electrode on polyimide (PI) substrate using pentacene as organic semiconductor channel to investigate the synaptic properties of the AiS-TSO (**Fig. 2a**).”

3. *Please indicate the thickness of the PVDF-TrFE/BC nanocomposite gate dielectrics.*

Response:

We added the cross-sectional FE-SEM image of BT NPs/P(VDF-TrFE) gate dielectric layer with the thickness (around 0.6 μm) in **Supplementary Figure 20c**.

a BT NP(20 wt%)/P(VDF-TrFE)

b BT NP(40 wt%)/P(VDF-TrFE)

c

Supplementary Figure 20. FE-SEM images of ferroelectric nanocomposites. FE-SEM images of ferroelectric nanocomposites of **a**, BT NP(20 wt%)/P(VDF-TrFE), **b**, BT NP(40 wt%)/P(VDF-TrFE) and **c**, cross-sectional FE-SEM image of the nanocomposite of BT NP(20 wt%)/P(VDF-TrFE) coated on Si wafer to confirm thickness of thin film layer. The thickness was estimated around 0.6 μm. The scale bar in all images is 1 μm.

We specified the thickness of gate dielectric layer and noticed the added image in the manuscript as follows (page 4):

“We first fabricated and characterized an Fe-OFET device with BT NP(20 wt%)/P(VDF-TrFE) nanocomposite gate dielectric (thickness of 0.6 μm) and Ni gate electrode on polyimide (PI) substrate using pentacene as organic semiconductor channel to investigate the synaptic properties of the AiS-TSO (**Fig. 2a**).”

In the method section, we also edited the sentence as follows: “**Dispersion of loaded BT NPs in the ferroelectric nanocomposite and the thickness of nanocomposite were examined by field-emission scanning electron microscopy (FE-SEM, JEO JSM-6500F).**”

4. Please indicate the magnitude of the coercive fields for Fe-OFETs with and without BT NPs, it is only stated that the composites have a smaller coercive field than pure PVDF-TrFE.

Response:

We appreciate the comments. We measured coercive fields of two samples with BT NP(20 wt%)/P(VDF-TrFE) nanocomposite and only P(VDF-TrFE) by fabricating structures in metal-ferroelectric-metal (MFM) on Si wafer with Pt bottom electrode and Al top electrode. To support the argument that the nanocomposite has a smaller coercive field than P(VDF-TrFE) in the main text, we added the polarization–electric field (P-E) curves in **Supplementary Figure 7**.

Supplementary Figure 7. Fundamental characteristics of polarization-electric field (P-E) curve of ferroelectric films. a, P-E curves of BT NP(20 wt%)/P(VDF-TrFE) and P(VDF-TrFE) thin films with the thickness of 1 μm . **b**, P-E curves of BT NP(20 wt%)/P(VDF-TrFE) nanocomposite thin film with the applied voltage varied. The sub-loops of P-E at the applied voltage range of -45 ~ +45 V and -60 ~ +60 V were also included. **c**, PSC (I_{PSC}) before and after applying the 100 pulses of V_{rec} (pulse width of 0.5 s and amplitude of -10 V) to monitor I_{PSC} related with the polarization.

To confirm the effect of BT NPs in the nanocomposite film on polarization behaviors, we fabricated the metal (Al)-ferroelectric-metal (Pt) (MFM) structures with the ferroelectric layers of BT NP(20 wt%)/P(VDF-TrFE) nanocomposite and P(VDF-TrFE) on Si wafer and measured polarization-electric field (P-E) curves by applying the voltage from -90 to 90 V (**Supplementary Fig. 7a**). The smaller coercive field (E_c) of ~39 MV/m for BT NP(20 wt%)/P(VDF-TrFE) nanocomposite film was obtained compared to that (~55 MV/m) for pure P(VDF-TrFE) film. Also, the nanocomposite film has much larger remnant

polarization (P_r) ($2.3 \mu\text{C}/\text{cm}^2$) than that of pure P(VDF-TrFE) ($0.3 \mu\text{C}/\text{cm}^2$). These results demonstrate that generation of larger P_r in BT NP (20 wt%)/P(VDF-TrFE) was observed compared to pure P(VDF-TrFE), which means that synaptic behaviors of the Fe-OFET can be tuned by varying the concentration of BT NPs in the nanocomposite. When the range of applied voltage was varied to smaller range, polarization also varied with a decreasing tendency with the applied voltage decreased. Therefore, polarization switching can be controlled according to the range of applied voltage⁸⁻¹², which implies that SW in Fe-OFET will depend on amplitude, duration time, rate and number of the V_{rec} pulses. To investigate the retention characteristics of PSC related with partial switching of dipoles, we measured the PSC with some time intervals after applying V_{rec} pulses of 100 times (pulse width of 0.5 s and amplitude of -10 V). It took more than 170 min to be fully recovered when the drain current was measured by applying the drain voltage of -1 V.

We added references 8-12 in Supplementary Information related to this explanation as follows:

8. Engel, S., Smykalla, D., Ploss, B. & Gräf, S. Effect of (Cd : Zn) S Particle concentration and photoexcitation on the electrical and ferroelectric properties of (Cd : Zn) S / P (VDF-TrFE) composite films. *Polymers* **9**, doi:10.3390/polym9120650
9. Xu, T., Xiang, L., Xu, M., Xie, W. & Wang, W. Excellent low-voltage operating flexible ferroelectric organic transistor nonvolatile memory with a sandwiching ultrathin ferroelectric film. *Sci. Rep.* **7**, 1–7 (2017). doi:10.1038/s41598-017-09533-2
10. Unni, K. N. N., Bettignies, R., Seignon, S.D. & Nunzi, J.M. A nonvolatile memory element based on an organic field-effect transistor. *Appl.Phys.Lett.* **85**, 1823–1825 (2004).
11. Yoon, S., Kim, E.J., Kim, Y.M. & Ishiwara, H. Adaptive-Learning Synaptic Devices using ferroelectric-gate field-effect transistors for neuromorphic applications. *2017 International Symposium on Nonlinear Theory and its Applications*, 311–333 doi:10.1007/978-94-024-0841-6
12. Park, B.E. *et al.* Ferroelectric- gate field effect transistor memories, *Springer*, **131** (2016). doi.org/10.1007/978-94-024-0841-6

Based on the P-E curves, we edited the explanation about synaptic properties related with polarization in the manuscript in page 6 as follows: “The larger, longer, and more repetitive was the applied V_{rec} , the larger was the polarization generated. Formation of LTP for the AiS-TSO with ferroelectric nanocomposite can be explained by the enhanced polarization switching at lower electric field, similarly to previous investigations of polarization switching dynamics of ferroelectric materials^{30–35}.”

Also, we mentioned the coercive field in page 7 as follows: “We found that the BT NP(20 wt%)/P(VDF-TrFE) nanocomposite has a smaller coercive field ($\sim 39 \text{ MV}/\text{m}$) than that of P(VDF-TrFE)

only (~55 MV/m), which implies enhanced dipole switching in the nanocomposite than in P(VDF-TrFE) (Supplementary Figure 7).”

5. A permanent polarization (which is claimed to be responsible for SW with LTP behavior due to a longer retention time of permanent polarization) can only occur, if the touch-induced voltage/potential is larger than the coercive field of the ferroelectric dielectric. In order to assess if this is really the case, it would be decisive to know the thickness of the pure PVDF-TrFE layer at least, which typically has a coercive field of ~ 50MV/m. Please provide more information on this. Furthermore, permanent polarization means that the polarization is permanent or, as it is mostly called, remanent and does not show any retention time.

Response:

The comments related with coercive field and permanent polarization are very appreciated. Through the analysis on P-E curves of BT NP(20 wt%)/P(VDF-TrFE) and P(VDF-TrFE) layers, we could get the smaller E_c and larger P_r for BT NP(20 wt%)/P(VDF-TrFE) layers compared to those of pure P(VDF-TrFE) (Supplementary Figure 7). As explained in the response for the comment #4, a larger remnant polarization (P_r) can be generated at a smaller applied voltage in BT NP(20wt %)/P(VDF-TrFE) film, which means the device using the nanocomposite ferroelectric gate dielectric layer can have larger SW and change of PSC than using those of the device with pure P(VDF-TrFE) (corresponding to the results of Fig.2j.-1. and 3g.-i). Also, since the SW is originated from the variation of polarization caused by partial switching of dipoles in ferroelectric gate dielectric layer below saturation, we agree with the comment on terminology “permanent polarization” which caused a confusion in meaning. Therefore, we edited the “permanent polarization” to “polarization” in the whole manuscript. Also, we added the results and discussion related with P-E curves, partial switching of dipoles and remanant polarization in Supplementary Figure 7, where detailed explanation was added (see the response for the comment #4). To investigate the retention time of the partial switched polarization, we measured PSC after applying 100 pulses of V_{rec} (pulse width of 0.5 s and amplitude of -10 V). When we measured PSC change with a time interval it took more than 170 min for PSC to be fully to recovered to the initial state.

Related with this result, we mentioned in manuscript in page 6 as follows: “And, the full recovery

time of I_{PSC} related with the retention of partially generated polarization after applying 100 pulses of V_{rec} (pulse width of 0.5 s and amplitude of -10 V) was about 170 min (**Supplementary Figure 7**).”

6. Could it be that the LTP behaviour, the increase of residual current or decay time with frequency (or number of pulses over a specific time period) is rather due to charge trapping in the OSC layer than to change in the “permanent” polarization?

Response:

Comments on influence of charge trapping of OSC layer are appreciated. To demonstrate that the LTP behavior occurred due to ferroelectric gate dielectric, not by charge trapping of OSC, we fabricated an OFET with the gate dielectric layers of non-ferroelectric PVP (polyvinylpyrrolidone) layer. We applied 1.42 Hz pulse (pulse width = 0.5s, amplitude = -10 V) on gate electrode and measured transfer curve. Also, when we compare the residual current right after the gate voltage removed, PVP OFET ($\sim 1 \times 10^{-11}$ A) was much smaller than that of Fe-OFET with the nanocomposite (BT NP(20 wt%)/P(VDF-TrFE)) ($\sim 1 \times 10^{-9}$ A). There was also an increase in the current in PVP OFET because of the charge trapping, but ferroelectricity of BT NPs/P(VDF-TrFE) in Fe-OFET has a greater effect on synaptic plasticity such as long-term potentiation. We added these data and explanation on **Supplementary Figure 3**.

Supplementary Figure 3. Comparison of characteristics OFET with ferroelectric and non-ferroelectric gate dielectric. a, Transfer curves of OFET with non-ferroelectric PVP (polyvinylpyrrolidone) gate dielectric and Fe-OFET with the ferroelectric nanocomposite gate dielectric (BT NP(20 wt%)/P(VDF-

TrFE)). **b**, Change of PSC (Δ PSC) in OFET with non-ferroelectric PVP gate dielectric and Fe-OFET with the ferroelectric nanocomposite gate dielectric (BT NP(20 wt%)/P(VDF-TrFE)) when the 1.42 Hz of pulses are applied on gate electrode (pulse width = 0.5 s, amplitude = -10 V).

To demonstrate that ferroelectric characteristics of gate dielectric layer in Fe-OFET mainly contribute to synaptic properties of the device, we also fabricated the OFET with non-ferroelectric PVP (polyvinylpyrrolidone) as a gate dielectric layer. The I_{PSC} of Fe-FET with nanocomposite showed larger hysteresis compared to that of OFET with PVP gate dielectric (**Supplementary Fig. 3a**). The change of PSC (Δ PSC) of the device with PVP ($\sim 1 \times 10^{-11}$ A) was much smaller than that of the device with BT NP(20 wt%)/P(VDF-TrFE) ($\sim 1 \times 10^{-9}$ A) (**Supplementary Fig. 3b**). Small Δ PSC of the device with PVP can be explained by the other effect such as charge trapping in organic semiconductor^{6,7}. Therefore, it can be argued that synaptic property in our Fe-OFET device is mainly attributed to polarization switching in ferroelectric gate dielectric layer via triboelectric-capacitive coupling.

Refs. 6 and 7 added on Supplementary Information as follows:

6. Tello, B. M., Chiesa, M., Duffy, C. M. & Siringhaus, H. Charge Trapping in Intergrain Regions of Pentacene Thin Film Transistors, *Adv.Funt.Mater.* **18**, 3907–3913 (2008).
7. Ha, R. & Batlogg, B. Gate bias stress in pentacene field-effect-transistors : Charge trapping in the dielectric or semiconductor. *Appl.Phys.Lett.* **99**, 083303 (2011).

And, we revised the manuscript to mention these further investigations in page 5 as follows: “Furthermore, to check the mechanism of synaptic weight generation in Fe-FET, we fabricated the OFET using non-ferroelectric polyvinylpyrrolidone (PVP) gate dielectric layer and characterized transfer curves and PSC by applying V_{rec} pulses of 1.42 Hz. SW in the OFET with PVP was negligible compared to that in the Fe-OFET with BT NP(20 wt%)/P(VDF-TrFE). Other mechanism such as charge trapping in organic semiconductor^{25,26} may not be enough to induce SW in the OFET with PVP (**Supplementary Fig. 3**). Therefore, it can be conjectured that generation of SW in the Fe-OFET is mainly attributed to ferroelectric polarization switching.”

Also, and added refs. 25 and 26 in manuscript as follows:

25. Tello, B. M., Chiesa, M., Duffy, C. M. & Sirringhaus, H. Charge Trapping in Intergrain Regions of Pentacene Thin Film Transistors, *Adv.Funt.Mater.* **18**, 3907–3913 (2008).
26. Ha, R. & Batlogg, B. Gate bias stress in pentacene field-effect-transistors : Charge trapping in the dielectric or semiconductor. *Appl.Phys.Lett.* **99**, 083303 (2011).

7. *It is unclear, how the intrinsic hysteresis effects in the Fe-OFETs, especially for the composites, will influence the PSC signals and their reproducibility (see SFig.1). In Fig.2f only the influence of increasing the Vrec amplitude (related to a claimed SADP) was investigated, but please show if this SADP effect is quantitatively reproducible also for decreasing the Vrec amplitude from -20V to -1V (especially for the composite material).*

Response:

The comments on the intrinsic hysteresis effects of the composites and reproducibility are appreciated. To demonstrate the repeatability of Fe-OFETs under electrical pulse, we characterized the device with gate pulse varied from -20 V to -1 V with decreasing the pulse amplitude. After the relaxation proceeds enough, there was not much difference in response and synaptic weight between the PSC under condition of increasing gate pulse amplitude (from -1 V to -20 V) or decreasing gate pulse amplitude (from -20 V to -1 V). These data and explanation were added on **Supplementary Figure 6**.

Supplementary Figure 6. Repeatability test with the full recovery of Fe-OFET. **a**, PSC when applying the increasing amplitude of V_{rec} from -1V to -20V with the full recovery of current. **b**, PSC when decreasing amplitude of V_{rec} from -20V to -1V with the full recovery of current. **c**, Merged PSC to demonstrate the repeatability of Fe-OFET. **c**, Compare the PSC with increasing and decreasing amplitude of V_{rec} . **d**, PSC when applying the increasing pulses number of V_{rec} from 1 to 100. **e**, PSC when applying the decreasing pulses number of V_{rec} from 100 to 1. **f**, Compare the PSC at 100 number of pulses applied in case of applying increasing and decreasing number of V_{rec} pulses. **g**, SW with increasing and decreasing of amplitude of V_{rec} with the full recovery. **h**, SW with increasing and decreasing of pulses number of V_{rec} with the full recovery.

To demonstrate the repeatability of Fe-OFET as a synaptic device, we applied the V_{rec} of increasing and decreasing amplitude of pulses and number of pulses. As shown in **Supplementary Figure 6**, PSC response was almost the same when we applied the V_{rec} in increasing or decreasing amplitude and pulse

number. Since we applied each pulse after full recovery to the state at the previous pulse, PSCs were only minimally affected by the previously formed polarization. Therefore, we found that if we need the repeatability of the device, we can apply the pulse after the full recovery. Here the repeatability in SW was good when we increase and decrease the amplitude of pulses and the number of pulses.

8. *Is the SNDP effect the same, if the number of spikes are decreased from e.g. 100 to 1 (Fig. 2g) ? This is important in order to judge the reproducibility and usefulness of the signals.*

Response:

We also characterized the device under the condition of decreasing the number of pulses and, similarly to the case of amplitude variation, the difference in responses of AiS-TSO under different measurement conditions of increasing and decreasing the number of pulses was not significant. These data were also added in **Supplementary Figure 6**.

Also, we added the sentence related with these results (comment # 7 and #8) in the manuscript in page 6 as follows: “Also, we measured the I_{PSC} of Fe-OFET by applying V_{rec} pulses with their amplitude and number consecutively increased or decreased after full recovery to confirm repeatability in the synaptic characteristics of Fe-OFET. The I_{PSC} values measured were almost similar when measured with increasing or decreasing amplitude or number of V_{rec} pulses (**Supplementary Figure 6**).”

9. *What about gate leakage currents in the Fe-OFETs?*

Response:

We measured and added the data of gate leakage current in **Supplementary Figure 1**.

Supplementary Figure 1. Basic characteristics of Fe-OFET and P-E curve of ferroelectric film. a, Output characteristics of Fe-OFET with P(VDF-TrFE) gate insulating layer. **b,** Output characteristics of Fe-OFET with BT NP(20wt %)/P(VDF-TrFE) nanocomposite gate insulating layer. **c,** Transfer curves of Fe-OFET with BT NP(20wt %)/P(VDF-TrFE) and P(VDF-TrFE). I_{PSC} and hysteresis are higher for Fe-OFET with BT NP(20wt %)/P(VDF-TrFE) compared to that with P(VDF-TrFE) only due to higher dielectric constant and lower coercive field. **d,** Gate leakage current of Fe-OFET with BT NP(20wt %)/P(VDF-TrFE) gate insulating layer.

Related with measurement of gate leakage current, we edited the sentence in manuscript as follows:

“Fundamental output characteristics, transfer characteristics and gate leakage current of the Fe-OFET are shown in **Supplementary Figure 1.**”

10. It has to be clearly stated that behaviour of the AiI-TSO under tensile strain was tested only for strains up to 1.88% which is far from what is the tensile strain strength of the skin in a human body (~ 25%, e.g. A.

Gallagher, IRC12-59, IRCOBI Conference). So the analogy of the proposed TSO with MCNCs is very limited in this regard. By the way, under repetitive compressive strain SFig 6e there is a large difference between initial values of Ipsc and compressed values.

Response:

The comment on the flexibility of the device is appreciated. We agree with some limitation of mimicking mechanical properties of the mechanoreceptor. In this work, we would like to put more value to adding more functional and morphological similarities that have not yet been implemented in mimicking mechanoreceptor devices.(refs. 14-20 in manuscript) Although our architecture is hard to keep up with the stable strain on the skin since it is flexible and not stretchable, it can be said that it is more advantageous to be used as an e-skin or soft robotics in terms of physical properties than a rigid device of silicon base. Therefore, our device has similar advantages with flexible devices having attracted attention and emerged as a candidate of e-skin or soft-robotics for many factors, including sensors mimicking mechanoreceptors, neuromorphic devices, and processors (Supplementary Information refs.11,13 and 17, and refers to doi.org/10.1002/adma.201903558). As the first work of mimicking intrinsic synaptic functions of mechanoreceptors and flexibility, we would like to regard our device as improved candidate for soft robotics and components of flexible AI systems. Also, accepting the comments, we would like to mention limit of current work and further improvement of our device for full utilization as e-skin by improving mechanical properties. Because our device is made of organic materials, there is a potential for an increase in conformality with thinner substrate or stretchable substrate with structural engineering for stress reduction. We revised manuscript to include these considerations as follow in page 7 as follows: “**AiS-TSO are not mechanically stretchable but flexible. There are some limitations in mimicking deformability of the skin for applications in electronic skin or soft robotics but its flexibility has many advantages compared to rigid devices. Further studies to improve stretchability are required to extend the capability of AiS-TSO.**”

Also, related with the **Supplementary Figure 9e**, we admit that there is some difference between PSC value of initial state and after compressive cyclic test. However, as shown in the graph, we could say that after applying a certain level of strain to the device, it is stabilized similarly to other flexible devices (Supplementary Information refs.13-17). The PSC after 10,000 cycles and 100,000 cycles did not make a

big difference, although the number of cycles increases more than that between the initial and 10,000 cycles. We also added this explanation to **Supplementary Figure 9** to clarify the phenomenon in details as follows: “The cyclic bending test results indicate that there is a significant difference in the I_{PSC} values between initial state and after compressive bending of 10,000 cycles (**Supplementary Fig. 9e**). The I_{PSC} values after 10,000 or 100,000 cycles are not much different even the number of bending cycles increases more than that between the initial and 10,000, which indicate that there is a stabilization stage similarly to other organic flexible devices^{13–17}.”

Refs. 13 to 17 added on Supplementary Information as follows:

13. Tien, N. T., Trung, T. Q., Seoul, Y. G., Kim, D. Il & Lee, N. Physically responsive field-effect Transistors with Giant Electromechanical Coupling Induced by Nanocomposite Gate Dielectrics. *ACS Nano* **5** 7069–7076 (2011).
14. Kim, D., Hwang, B., Tien, N. T., Kim, I. & Lee, N. Effects of piezoresistivity of pentacene channel in organic thin film transistors under mechanical bending. *Electron. Mater. Lett.* **8**, 11–16 (2012).
15. Kim, Y. *et al.* Flexible textile-based organic transistors using Graphene / Ag nanoparticle electrode. *Nanomaterials* **6**, (2016). doi:10.3390/nano6080147
16. Ding, Y. *et al.* Flexible small-channel thin-film transistors by electrohydrodynamic lithography. *Nanoscale* **9**, 19050-19057 (2017).
17. Liu, C., Fujimoto, Y. & Tanaka, Y. Flexible impact force sensor. *J.S.T.* **4**, 66–80 (2014).

11. Please explain, how the claimed tunable filtering ability in terms of SW can be implemented in a real-world artificial TSO, since it is mainly based on using ferroelectric materials with different NP concentration.

Response:

Comment on the consideration of implementing the filtering ability in terms of SW is appreciated. First, we would like to explain our tunable filtering ability by comparison with mechanoreceptors in our body. The mechanoreceptors in our body differ in the nature and extent of the stimuli they detect (frequency, strength etc.). (refs. 21-24 in manuscript) This is because the characteristics of each mechanoreceptor and the structure of connection between afferent neuron are different. In other words, there are types and ranges of stimuli that are received and delivered depending on the mechanical receptor, and others are filtered. (refs. 12,13, and 21-24 in manuscript) As mentioned in the manuscript (page 2,3 and 8), Merkel cell neurite

complex (MCNC) also plays a role in detecting and filtering specific stimuli with synapse-like connection (refs. 21-24 in manuscript). By the way, it is obvious that neurons and synapse in our brain filter the accepted information depends on synaptic weight (refs.5-10 in manuscript). Mimicking these filtering functions of our tactile perception system, especially, filtering of MCNC with synapse-like connection, we tried to design the device that has a capability of SW generation. Our device can filter the noise signals intrinsically through SW as other synaptic device and our brain do. Therefore, SW in a single AiS-TSO is modulated by the degree of input stimuli.

Secondly, we tried to demonstrate that the different levels of SW generated for the same stimuli in the AiS-TSO with the nanocomposite with different BT NPs loading having varied ferroelectric properties, the criteria of noise filtering in device itself can be set differently. For example, when filtering the stimuli using the same SW criteria, the device with the nanocomposite of 40 wt% BT NPs will accept a low stimulus in the range that is not acceptable by filtering with noise for the device with the nanocomposite of 20wt% BT NPS. In other words, the threshold for noise filtering can be set differently by using the different nanocomposite in different AiS-TSOs. Through this concept, we presume that AiS-TSOs with different thresholds for noise filtering may simply setting up of the filtering criteria of each device. These days, there have been a lot of efforts on the analysis and mimicking filtering functions of mechanoreceptors (refers to DOI: 10.1109/IJCNN.2008.4633797, 10.1162/089976604773135069, 10.1016/j.neuron.2017.09.004 and 10.1162/089976604773135069) since these filtering functions are essential to artificial tactile perception system for neuromorphic e-skin, neuro-robotics, brain-machine interface and components of AI system. Therefore, we believe that tuneability of synaptic properties in AiS-TSO may provide great potential to be applied to a more sophisticated tactile perception system since 1) it has inherent filtering function, and 2) its filtering function is also hardware-adjustable.

Related with this discussion, we tried to clarify filtering functions of AiS-TSO and revised the manuscript as follows: “Using different SW change depending on the concentration of BT NPs in the nanocomposites, therefore, we could set the different criteria to be used for noise filtering or getting specific range of information similarly to biological mechanoreceptors which transfer signals to brain depending on characteristics of the receptor cells or number and distribution of connections between afferent neurons¹².”

A reference as added related with functions of Merkel-cell neurite complex as follows:

12. Hu, J., Zhao, Q., Jiang, R., Wang, R. & Ding, X. Responses of cutaneous mechanoreceptors within fingerpad to stimulus information for tactile softness sensation of materials. *Cogn. Neurodyn.* **7**, 441–447 (2013).

12. In Fig.4e it is not clear why it can be deduced that the memory strength of Pixel 3 of 1.3 means clearly that the Pixel 3 was not touched, whereas a strength of 1.6 for Pixel 2 means that the Pixel 2 is touched. The difference seems to be too small to make an unambiguous decision.

Response:

We appreciate the comments on the ambiguous different between the firstly touched pixel and non-touched pixel. To demonstrate such an application as memory embedded functions, we conducted the experiment of memory application in the case of $2 \rightarrow 1 \rightarrow 4 \rightarrow (3)$ three times, and, analyzed memory strength. We found that the devices have difference in memory strength depending on the order of touch. And, three repetitive experiments show the same tendency that non-touched device has the smallest memory value and memory strength increased from the firstly touched (pixel 2) to the lastly touched device (pixel 4). The data obtained from repetitive measurements indicate that the memory stored in the devices touched sequentially enable us to distinguish the order of touch reliably. The data were added in **Supplementary Figure 22** as follows:

Supplementary Figure 22. Measurement of memory strength to demonstrate memory-embedded function in AiS-TSO upon finger touches.

Here, we measured the memory strength of AiS-TSO in an order of touch, pixel 2→1→4→(3) while the pixel 3 was skipped (not touched). The measurements were repeated three times. All three experiments showed the same tendency that the untouched pixel 3 has the smallest memory strength and the memory strength was increased from the firstly touched (pixel 2) to lastly touched device (pixel 4).

Also, we added a sentence in the manuscript related with these extra experiments as follows in page 12: “To confirm that memory strength is related to the order of touch, originated from SW and retention time, we measured the memory strength three times with the touch order of pixel 2→1→4 without touching of pixel 3 (**Supplementary Figure 22**). We could find that the memory strength has the same tendency for all three measurements, which is consistent with the results in **Fig. 4e**.”

13. Please remove the term “self-powered” since the device clearly needs the application of a potential to the drain electrode, in order to be functional.

Response:

We appreciate the comment on the term of “self-powered”. We used this term to express that our device needs no external electrical energy to induce dipole switching in ferroelectric material. Similar to mechanoreceptors in our body, converting the mechanical energy into action potential by transferring the ions and neurotransmitters could be matched with the concept of our device in which the mechanical energy switches dipoles and is converted to electric energy by transferred triboelectric charges. However, we agree with the comment that AiS-TSO is hard to think of fully “self-powered” because we need read voltage at drain electrode. Although AiS-TSO is not totally self-powered, it is obvious that the device generates the electrical energy by stimulation and so we would like to change the term of “self-powered” to “self-energy transducing” to emphasize the functions of energy generating and its similarity to function of mechanoreceptors. Therefore, we revised the main text as follows (page3): “Touch stimulation induces alignment of dipoles in the ferroelectric gate dielectric by triboelectric-capacitive coupling effect which causes the post-synaptic current signal to be modulated, thereby allowing tactile information to be imparted to the signal in a self-energy transducing manner. The synaptic function of the device enables the output signal to be pre-processed through the multiple functions of slow adaption (SA), filtering and memory in a self-energy transducer manner.”

Also, we edited the caption of **Fig. 1b** as follows: Reception by a flexible AiS-TSO occurs in a self-energy transducing way by triboelectrification during finger touch.

14. The influence of repetitive pulses or number of pulses on the I_{psc} (SNDP) or paired pulses was not investigated for the AiI-TSO. What is expected here?

Response:

The comment on insufficient data from AiS-TSO related with repetitive pulses, I_{PSC} with the number of pulses and paired pulses ratio (PPR). We already presented the effect of repetitive electrical pulses in the main text. We presume that the reviewer commented on repetitive touch experiment with different frequency. Per reviewer’s comment, we further investigated the effect of repetitive touch and characterized the behavior of the device. First, we conducted the experiment on repetitive touch. We measured the PSC with repetitive touch at the condition of touch for 1s and relaxation for 5 s and 10 s. We could observe the tendency of

increasing PSC with repetitive touch and the PSC change ratio and SW were larger in case of shorter relaxation after touch (5s). Next, we added the data related to SNDP and obtained the PPR depending on interval time between two consecutive touches. We added the PSC and SWs depending on number of touches were shown in **Fig. 4b** in manuscript. PPR was increased when the time interval between two touches was decreased.

Newly acquired data were added in **Supplementary Figure 18** as below. Also, we discussed these results in the revised manuscript as follows in page 10:

“We further investigated the characteristics of AiS-TSO such as PSC change ratio ($\Delta I_{PSC}/I_{PSC,i}$) by repetitive touch with different retention time and PPR with different time interval of touch(**Supplementary Fig. 18**). We observed that the shorter retention time made the $\Delta I_{PSC}/I_{PSC,i}$ larger. Also, PPR was increased when the time interval of two consecutive touches was decreased.

Supplementary Figure 18. Characteristics of AiS-TSO with repetitive touches and different number of touches. **a**, Change ratio of PSC ($\Delta I_{PSC}/I_{PSC,i}$) with repetitive touches having retention time of 1 s and retention time of 5s and 10 s. **b**, I_{PSC} with consecutive touches depending on different time interval and PPR value. All the data were obtained at the touch force of ≈ 1 kPa.

Related with SNDP in AiS-TSO data were added in **Supplementary Figure 21** as below. Also, we added the explanation in the revised manuscript as follows in page 12: “Therefore, SNDP in AiS-TSO observed with finger touch (**Fig. 4c and Supplementary Figure 21**) shows potential to be used as a sensing

device embedded with memory function at unit device level without the need for an additional memory device.”

Supplementary Figure 21. I_{PSC} with the number of touches varied. Touch force of ≈ 1 kPa varying number of touches.

Reviewer #3:

1. This paper reports a sensor that utilizes a ferroelectric composite as dielectric transistor material.

In general, the wording used in this manuscript is often times complicated, difficult to read, due to punctuation errors, and misleading.

Some examples are:

Title: “A Flexible Artificial Intrinsic-Intelligent Tactile Sensory Organ”, which claims an “intelligent” sensory organ, whereby it is completely unclear how the demonstrated sensor is “intelligent”, i.e. has the ability to acquire and apply knowledge and skills (from Oxford Dictionary);

Response:

Comment on the use of confusing terms is greatly appreciated. We agree that comments of “intelligent” could be used to the device which can acquire and apply the knowledge. Here we used the “intelligent” term since we regarded the functions of adaptation, memory and filtering as necessary to conduct intelligent process both in our body and AI device. Even though our device has limitation to be regarded as device acquiring or applying knowledge, it can provide the pretreated data to conduct intelligent process without any extra device so that it is potential to be part of further intelligent system. As mentioned in manuscript, functions (adaptation, filtering and memory) inherent in the device itself, which have been regarded as synaptic functions in brain and these are originated to synapse-like connections of sensory organs. Therefore, taking the comment and expressing the novelty of our device which have intrinsic synaptic functions at first time, we would like to change our title and device name as follow: “**A Flexible Artificial Intrinsic-Synaptic Tactile Sensory Organ**” and “**AiS-TSO**”.

Also, the sentence in the abstract was revised in page 2 as follows: “**Here, inspired by structure and functions of Merkel cells which form synapse-like connections with afferent neuron terminals, referred to as Merkel cell-neurite complexes (MCNCs), we report a flexible, artificial, intrinsic-synaptic tactile sensory organ (AiS-TSO) that mimics synapse-like connections using an organic synaptic transistor with ferroelectric nanocomposite gate dielectric of barium titanate nanoparticles and poly(vinylidene fluoride-trifluoroethylene).**”

And, the sentence in page 3 in manuscript also were edited as follows: “**Herein, we demonstrate a flexible, artificial, intrinsic-synaptic tactile sensory organ (AiS-TSO) that mimics the synapse-like connections of MCNCs, conferring intrinsic synaptic properties to the unit sensor for conducting further intelligent work.**”

-or “*Sensory organs enable animals to gather information to conduct skilled movements...*”, where sensors are linked to skilled movements, while one has nothing to do with the other;

Response:

Comment on the use of confusing sentence is appreciated. Agreeing with that comment, we edited

the sentence so that it has more clear meaning. To convey meaning of the sentence clearly, we edited the sentence in page 2 as follows: “Sensory organs enable animals to gather information for perception and to have ability for lives such as conducting skilled movements and seeking protection from hazardous situations.”

- or “Even though much effort has been made to mimic mechanoreceptors in the human body, emulation of their intelligent functions to extend sensory reception in an efficient way has not been widely successful. ”, which is a generic statement that lacks explanation what “in an efficient way” means and where “... has not been widely successful” is also a completely vague statement.

Response:

We appreciated the comment on vague meaning of the sentence. To clarify the meaning of the sentence, we changed the sentence in page 2 as follows: “Even though much effort has been made to mimic mechanoreceptors in the human body¹⁴⁻¹⁸, emulation of their intelligent functions to extend sensory reception has not been widely successful due to difficulty in adding synaptic connection between receptors and neuron terminal at the device level.”

-The meaning of sentences like “Touch stimuli control alignment of permanent dipoles in the ferroelectric gate dielectric so that the output signal is endowed with tactile information with the parallel functions of slowly adapting (SA) sensation, filtering, and memory in a self-powered manner. ” is obviously difficult to understand, due to missing words etc.

Response:

We appreciate the comment on the sentence which has difficulty in understanding. To make the sentence better understood, we edited the sentence in page 3 as follows: “Touch stimulation induces alignment of dipoles in the ferroelectric gate dielectric by triboelectric-capacitive coupling effect which causes the post-synaptic current signal to be modulated, thereby allowing tactile information to be imparted to the signal in a self-energy transducing manner. The synaptic function of the device enables the output

signal to be pre-processed through the multiple functions of slow adaption (SA), filtering and memory in a self-energy transducer manner.”

2. Characterization of the ferroelectric composite is absent. The minimum should be a polarization curve (P over E) to see the remanent polarization and permittivity of the material.

Has the composite been exposed to any poling? Otherwise, the dipole alignment would be random.

Hysteresis measurements of the polarization for the applied electric field would be very interesting too, in order to understand the hysteresis effect on the gate potential.

Response:

The comment on the missing data related with characterization of ferroelectric nanocomposite is appreciated. We added the polarization vs. electric field (P-E) curves for pure P(VDF-TrFE) and BT NP(20 wt%)/P(VDF-TrFE) nanocomposite by fabricating metal-insulating-metal structure on Si wafer. The remanant polarization (P_r) of BT NP(20 wt%)/P(VDF-TrFE) nanocomposite was larger than that of pure P(VDF-TrFE). We added the data and explanation in **Supplementary Figure 7** as follows:

Supplementary Figure 7. Fundamental characteristics of polarization-electric field (P-E) curve of ferroelectric films. a, P-E curves of BT NP(20 wt%)/P(VDF-TrFE) and P(VDF-TrFE) thin films with the thickness of 1 μm . **b,** P-E curves of BT NP(20 wt%)/P(VDF-TrFE) nanocomposite thin film with the applied voltage varied. The sub-loops of P-E at the applied voltage range of -45 ~ +45 V and -60 ~ +60 V were also included. **c,** PSC (I_{PSC}) before and after applying the 100 pulses of V_{rec} (pulse width of 0.5 s and amplitude of -10 V) to monitor I_{PSC} related with the polarization.

To confirm the effect of BT NPs in the nanocomposite film on polarization behaviors, we fabricated the metal (Al)-ferroelectric-metal (Pt) (MFM) structures with the ferroelectric layers of BT NP(20 wt%)/P(VDF-TrFE) nanocomposite and P(VDF-TrFE) on Si wafer and measured polarization-electric field (P-E) curves by applying the voltage from -90 to 90 V (**Supplementary Fig. 7a**). The smaller coercive field (E_c) of ~39 MV/m for BT NP(20 wt%)/P(VDF-TrFE) nanocomposite film was obtained compared to that (~55 MV/m) for pure P(VDF-TrFE) film. Also, the nanocomposite film has much larger remnant

polarization (P_r) ($2.3 \mu\text{C}/\text{cm}^2$) than that of pure P(VDF-TrFE) ($0.3 \mu\text{C}/\text{cm}^2$). These results demonstrate that generation of larger P_r in BT NP (20 wt%)/P(VDF-TrFE) was observed compared to pure P(VDF-TrFE), which means that synaptic behaviors of the Fe-OFET can be tuned by varying the concentration of BT NPs in the nanocomposite. When the range of applied voltage was varied to smaller range, polarization also varied with a decreasing tendency with the applied voltage decreased. Therefore, polarization switching can be controlled according to the range of applied voltage⁸⁻¹², which implies that SW in Fe-OFET will depend on amplitude, duration time, rate and number of the V_{rec} pulses. To investigate the retention characteristics of PSC related with partial switching of dipoles, we measured the PSC with some time intervals after applying V_{rec} pulses of 100 times (pulse width of 0.5 s and amplitude of -10 V). It took more than 170 min to be fully recovered when the drain current was measured by applying the drain voltage of -1 V.

We added references 8-12 in Supplementary Information related to this explanation as follows:

8. Engel, S., Smykalla, D., Ploss, B. & Gräf, S. Effect of (Cd : Zn) S Particle concentration and photoexcitation on the electrical and ferroelectric properties of (Cd : Zn) S / P (VDF-TrFE) composite films. *Polymers* **9**, doi:10.3390/polym9120650
9. Xu, T., Xiang, L., Xu, M., Xie, W. & Wang, W. Excellent low-voltage operating flexible ferroelectric organic transistor nonvolatile memory with a sandwiching ultrathin ferroelectric film. *Sci. Rep.* **7**, 1–7 (2017). doi:10.1038/s41598-017-09533-2
10. Unni, K. N. N., Bettignies, R., Seignon, S.D. & Nunzi, J.M. A nonvolatile memory element based on an organic field-effect transistor. *Appl.Phys.Lett.* **85**, 1823–1825 (2004).
11. Yoon, S., Kim, E.J., Kim, Y.M. & Ishiwara, H. Adaptive-Learning Synaptic Devices using ferroelectric-gate field-effect transistors for neuromorphic applications. *2017 International Symposium on Nonlinear Theory and its Applications*, 311–333 doi:10.1007/978-94-024-0841-6
12. Park, B.E. *et al.* Ferroelectric- gate field effect transistor memories, *Springer*, **131** (2016). doi.org/10.1007/978-94-024-0841-6

Based on the P-E curves, we edited the explanation about synaptic properties related with polarization in the manuscript in page 6 as follows: “The larger, longer, and more repetitive was the applied V_{rec} , the larger was the polarization generated. Formation of LTP for the AiS-TSO with ferroelectric nanocomposite can be explained by the enhanced polarization switching at lower electric field, similarly to previous investigations of polarization switching dynamics of ferroelectric materials^{30–35}.”

Also, we mentioned the coercive field in page 7 as follows: “We found that the BT NP(20 wt%)/P(VDF-TrFE) nanocomposite has a smaller coercive field ($\sim 39 \text{ MV}/\text{m}$) than that of P(VDF-TrFE)

only (~55 MV/m), which implies enhanced dipole switching in the nanocomposite than in P(VDF-TrFE) (Supplementary Fig. 7).”

3. The described concept has previously been called POSFET touch sensitive devices that use piezoelectric polymers for tactile sensing. Therefore, the novelty of this work is not obvious. Especially the title implies a new concept, which, does not seem to be justified.

Response:

We appreciate the comment on novelty of our device from POSFET. We can explain surely that there are obvious differences between our device and POSFET and from that our device has own novelty. To make sure that AiS-TSO has different materials, structure and functions from POSFET, we added additional figure and detailed explanation in **Supplementary Figure 16** as follows:

Supplementary Figure 16. Comparison between piezoelectric oxide semiconductor FET (POSFET) and AiS-TSO as touch sensor. Schematic illustration of working principle of **a**, POSFET tactile sensor and **b**, AiS-TSO during touch (left) and after touch (right).

Firstly, working principle of POSFET is much different from that of AiS-TSO. POSFET touch sensor utilizes the piezoelectric response of the piezoelectric gate dielectric while AiS-TSO does not utilize

the piezoelectric effect of the gate dielectric. In POSFET, touch stimuli induce i) the displacement of polarized piezoelectric material and ii) change the electric field in piezoelectric material, which modulates the carrier density in the channel and, in turn, the drain current (**Supplementary Fig. 16a**). Here, the piezoelectric layer upon pressurizing induces change in dipole alignment resulting in change in effective gate electric field²⁸⁻³⁰. Therefore, POSFET needs intended polling process for generating the saturated remanant polarization, P_r , as much as possible to generate piezoelectric voltage enough to modulate the drain current. In AiS-TSO, on the other hand, i) triboelectrification between skin and polyimide substrate (described as receptive part in the manuscript) generates triboelectric charges and ii) coupled capacitive effect in the receptive part induces the partial dipole switching in ferroelectric material and, in turn, change in the drain current (**Supplementary Fig. 16b**). Therefore, mechanisms of generating and transduction of energy are different, in which the POSFET sensor uses piezoelectric effect while our AiS-TSO uses triboelectric-capacitive coupling effect between skin and receptive part.

Secondly, the functions of sensor have differences. Both sensors have a common function with energy transducer from mechanical to electrical as a device of mimicking mechanoreceptors, but AiS-TSO adds intrinsic synaptic functions and enhances the functionalities for mimicking mechanoreceptors (mimicking synaptic functions of Merkel cell neurite complex). Differently from POSFET, we do not carry out intended polling process to generate P_r in ferroelectric material. Instead, we induce modulation of polarization with touch stimuli causing dipole switching depending on nature of stimuli. Therefore, due to the characteristics of ferroelectric material, the conductivity changes in the channel changes are inherently endowed with information of touch stimuli. In conclusion, we could induce SW through the modulation of polarization switching in ferroelectric layer under varying stimuli of touch resulting in inherent change in post-synaptic current and, in turn, modulation of SW. Thus, AiS-TSO has the advantages of a simple structure and manufacturing process, and unlike other mechanoreceptor-mimetic sensors whose only detection function has been reported, AiS-TSO has an intrinsic synaptic function that mimics the Merkel cell neurite complex.

The references 28-30 of Supplementary Information are added as follows:

28. Dahiya, R. S., Lorenzelli, L., Metta, G. & Valle, M. POSFET devices based tactile sensing arrays. *Proc. 2010 IEEE Int. Symp. Circuits Syst.* 893–896 (2010). doi:10.1109/ISCAS.2010.5537414

29. Adami, A., Dahiya, R. S., Collini, C., Cattin, D. & Lorenzelli, L. POSFET touch sensor with CMOS integrated signal conditioning electronics. *Sensors Actuators A. Phys.* **188**, 75–81 (2012).
30. Dahiya, R. S., Adami, A., Collini, C. & Lorenzelli, L. POSFET tactile sensing arrays using CMOS technology. *Sensors Actuators A. Phys.* **202**, 226–232 (2013).

For revised manuscript, we added the sentence in page 9 as follows: “In addition, functions of AiS-TSO are clearly different from those of conventional piezoelectrically-coupled tactile sensor such as piezoelectric oxide semiconductor FET (POSFET)^{47–49}. While our AiS-TSO utilizes polarization switching in ferroelectric gate dielectrics through the energy transducing mechanism of triboelectric-capacitive coupling, POSFET utilizes alignment change of well-aligned dipoles in pre-poled piezoelectric gate dielectrics through piezoelectric coupling. More detailed explanation is given in supplementary information (**Supplementary Figure 16**).”

The references 47-49 which are related with this discussion also added as follows:

47. Dahiya, R. S., Lorenzelli, L., Metta, G. & Valle, M. POSFET devices based tactile sensing arrays. *Proc. 2010 IEEE Int. Symp. Circuits Syst.* 893–896 (2010). doi:10.1109/ISCAS.2010.5537414
48. Adami, A., Dahiya, R. S., Collini, C., Cattin, D. & Lorenzelli, L. POSFET touch sensor with CMOS integrated signal conditioning electronics. *Sensors Actuators A. Phys.* **188**, 75–81 (2012).
49. Dahiya, R. S., Adami, A., Collini, C. & Lorenzelli, L. POSFET tactile sensing arrays using CMOS technology. *Sensors Actuators A. Phys.* **202**, 226–232 (2013).

4. *The authors show very well the ability to adjust the synaptic weight by varying the filler content. However, using this method the weight is defined during the fabrication and no changes can be made afterwards, compared to the real synapses that have the ability to strengthen or weaken the weight. This synaptic plasticity is a crucial ability for learning and adaptation.*

Response:

Related with the comment, we would like to explain about the control of synaptic weight (SW) in our device as we described in manuscript. Firstly, all AiS-TSOs with different ferroelectric nanocomposites have synaptic properties and their SW is modulated under different electric pulses and stimuli. Secondly, in addition, we tried to show that we can fabricate AiS-TSOs with different degree of synaptic properties by varying the composition of the nanocomposite having different ferroelectric properties. Using the different

ferroelectric nanocomposites, we can provide the tunability of threshold for noise filtering in AiS-TSO as an application.

As we discussed in the main text, we could control the SW in each AiS-TSO by varying the gate voltage which is expressed as receptor potential, V_{rec} , and different touch stimuli. As shown in the **Fig. 2** in the manuscript, we applied the V_{rec} with varying the amplitude, frequency, duration time and number of pulses and explanation on the effects were described in page 5 and 6 of the manuscript. Also, we could modulate the SW like strengthening or weakening with negative or positive V_{rec} . These data are described in **Supplementary Figure 4, 5 and 8** in more detail with the analysis of SW, PPR and retention time. It was shown that SW of AiS-TSO can be modulated with different nature of tactile stimuli (force, duration time, frequency and number of touch) (**Fig. 3 and 4**).

Once AiS-TSO is made with a nanocomposite with a specific composition BT NPs, its synaptic property is fixed. Therefore, degree of SW modulation under the same stimuli is different in AiS-TSO devices with the different nanocomposites. In summary, our point was that synaptic properties of ferroelectric materials with different concentration of BT NPs in AiS-TSO could be tuned while all AiS-TSP devices with different nanocomposite has their own synaptic property and SW modulation by different nature of touch stimuli.

Related with this discussion, we tried to clarify filtering functions of AiS-TSO so that we revised the manuscript as follows: “Using different SW change depending on the concentration of BT NPs in the nanocomposites, therefore, we could set the different criteria to be used for noise filtering or getting specific range of information similarly to biological mechanoreceptors which transfer signals to brain depending on characteristics of the receptor cells or number and distribution of connections between afferent neurons¹².”

A reference was added related with functions of Merkel-cell neurite complex as follows:

12. Hu, J., Zhao, Q., Jiang, R., Wang, R. & Ding, X. Responses of cutaneous mechanoreceptors within fingerpad to stimulus information for tactile softness sensation of materials. *Cogn. Neurodyn.* **7**, 441–447 (2013).

5. On the other hand, it seems that the synaptic weight could also be adjusted by adjusting the gate voltage,

as shown in this paper, a parameter that is much easier to control in an artificial system, and which could be adjusted over time in a learning-like manner. An “imprinted” synaptic weight could also be achieved during fabrication by other means like channel width.

Response:

We appreciate the comments on parameters of adjusting the synaptic weight (SW). Following the comment, we considered the effect of the channel length on the synaptic property by varying it from 40 μm to 70 μm . As the channel length increases, the initial value of I_{PSC} increases following the relation of drain current and channel length in field-effect transistor (ref. 13 in Supplementary Information). Also, we could find out that the synaptic weight is affected by channel length. SW values were increased with the channel length increased (**Supplementary Fig. 10b**). Therefore, we added explanation and the data related with the “imprinted” synaptic phenomenon in **Supplementary Figure 10** as follows:

Supplementary Figure 10. Characteristics of Fe-OFET depending on channel length. a, I_{PSC} with varying the number of V_{rec} pulses from 1 to 50 from Fe-OFETs with channel length of 40, 50 and 70 μm (pulse width of 0.2 s and amplitude of -10 V). **b,** synaptic weight (SW) analysis depending on the channel length and number of V_{rec} pulses.

To investigate the channel length effect of Fe-OFET, we fabricated the device with different channel lengths of 40, 50 and 70 μm . As shown in **Supplementary Figure 10a**, the I_{PSC} level was increased with the

channel length decreased, as expected. On the other hand, as shown in **Supplementary Figure 10b**, SW increases as the channel length increases. This observation can be explained as an effect of increased retention time with the channel length increased which is related with the gate area of Fe-OFET^{10,12,18–21} and slower switching time of dipoles in ferroelectric gate dielectric of Fe-OFET^{12,20,22}

Related with this explanation, we added references in Supplementary Information as follows:

10. Unni, K. N. N., Bettignies, R., Seignon, S.D. & Nunzi, J.M. A nonvolatile memory element based on an organic field-effect transistor. *Appl.Phys.Lett.* **85**, 1823–1825 (2004).
12. Park, B.E. *et al.* Ferroelectric- gate field effect transistor memories, *Springer*, **131** (2016). doi.org/10.1007/978-94-024-0841-6
18. Yurchuk, E. *et al.* HfO₂-based ferroelectric field-effect transistors with 260 nm channel length and long data retention. (2012). doi:10.1109/IMW.2012.6213620.
19. Yurchuk, E. *et al.* Impact of scaling on the performance of HfO₂-based ferroelectric field effect transistors. *IEEE*. **61**, 3699–3706 (2014).
20. Muller, J. *et al.* Nanosecond polarization switching and long retention in a novel MFIS-FET Based on Ferroelectric HfO₂. *IEEE*. **33**, 185–187 (2012).
21. Gating, Z. F. *et al.* MoS₂ field-effect transistors with Lead Zirconate-Titanate ferroelectric gating. *IEEE*. **36**, 784–786 (2015).
22. Sugano, R. *et al.* Switching Time in Ferroelectric Organic Field-Effect Transistors. *Adv.Sci.* **215**, 1701059 (2018).

We added a sentence in the manuscript(page8) related with effect of channel length modulation for synaptic weight controlling as follows: “One of factors affecting SW is the device scaling which includes changes in the thickness of ferroelectric layer and channel length or width^{37–42} in Fe-FET. Decrease in the channel length of our Fe-OFET enhanced I_{PSC} (**Supplementary Figure 10**). However, increase in the channel length enhanced SW due to larger retention time which might be related to slower polarization switching^{38,40,43}.”

Related with this mention, we also added the reference 37-43 in manuscript as follows:

37. Yurchuk, E. *et al.* HfO₂-based ferroelectric field-effect transistors with 260 nm channel length and long data retention. *IEEE*. (2012). doi:10.1109/IMW.2012.6213620.
38. Park, B.E. *et al.* Ferroelectric- gate field effect transistor memories, *Springer*, **131** (2016). doi.org/10.1007/978-94-024-0841-6
39. Yurchuk, E. *et al.* Impact of scaling on the performance of HfO₂-based ferroelectric field effect transistors. *IEEE*. **61**, 3699–3706 (2014).
40. Muller, J. *et al.* Nanosecond polarization switching and long retention in a novel MFIS-FET Based on

Ferroelectric HfO₂. *IEEE*. **33**, 185–187 (2012).

41. Gating, Z. F. *et al.* MoS₂ field-effect transistors with Lead Zirconate-Titanate ferroelectric gating. *IEEE*. **36**, 784–786 (2015).
42. Unni, K. N. N., Bettignies, R., Seignon, S.D. & Nunzi, J.M. A nonvolatile memory element based on an organic field-effect transistor. *Appl.Phys.Lett.* **85**, 1823–1825 (2004).
43. Sugano, R. *et al.* Switching Time in Ferroelectric Organic Field-Effect Transistors. *Adv.Sci.* **215**, 1701059 (2018).

6. *An interesting issue of the proposed approach is the practicality when considering array fabrication, where I imagine it very difficult to adjust the filler concentration from one transistor to another on the same wafer, in order to obtain the desired weights.*

Response:

Comment on issues of the practical usage of materials with adjusting the filler concentration is appreciated. The concern is about the adjustment of BT NPs concentration in each device for array of devices with different BT NP concentrations. In fact, array of devices with different concentration might be needed. Since biological mechanoreceptors also have filtering function as we described, depending on the shape, characteristic of receptor cells and distribution of connections with afferent neurons.(refs.12-13 and 21-24 in manuscript) Therefore, for advanced mimicking of mechanoreceptors for applications as e-skin or neuro-robotics, there are attracted attentions on analysis of neural signals related with filtering functions of mechanoreceptors and implementation of the filtering functions on bio-mimetic sensors. In this work, we tried to implement the filtering functions in the device itself, without any extra device, intrinsically by using intrinsic SW formation and control. Furthermore, we could modulate the degree of SW generation through control of the concentration of BT NPs in ferroelectric nanocomposite. Therefore, we could broaden the selection of thresholds for noise filtering criteria. For example, when filtering the stimuli using the same SW criteria, the device with the nanocomposite of 40 wt% BT NPs will accept a low stimulus in the range that is not acceptable by filtering with noise for the device with the nanocomposite of 20wt% BT NPs. In other words, the threshold for noise filtering can be set differently by using the different nanocomposite in different AiS-TSOs. In this work, we fabricated the AiS-TSO with the same concentration for four devices on single substrate as a simple demonstration of advanced artificial mechanoreceptors which have filtering function and, furthermore, control of the function.

However, we agree with the issues on practical uses of this concept. For practical uses, we should fabricate the devices with different BT NPs concentration on the same substrate to have an array like biological mechanoreceptors which detect different range of stimuli and generate the signals through stimuli filtering functions. Since we used spin coating method for formation of BT NPs/P(VDF-TrFE) film in this work, it could have limitations for patterning the nanocomposite with different BT NPs/P(VDF-TrFE) on the same substrate in a large scale with high uniformity. Therefore, we can use various printing methods for nanocomposites patterned with different concentration of BT NPs and of uniform large-scale device array for tactile perception system in e-skin or neuro-robotics mimicking human skin which is the largest organ of human. We should consider the improvement in developments of solutions of the nanocomposite with better dispersion control and development of additive printing methods.

We edited these explanation and discussion in the manuscript as follows: “Using different SW change depending on the concentration of BT NPs in the nanocomposites, therefore, we could set the different criteria to be used for noise filtering or getting specific range of information similarly to biological mechanoreceptors which transfer signals to brain depending on characteristics of the receptor cells or number and distribution of connections between afferent neurons¹².”

A reference was added related with functions of Merkel-cell neurite complex as follows:

12. Hu, J., Zhao, Q., Jiang, R., Wang, R. & Ding, X. Responses of cutaneous mechanoreceptors within fingerpad to stimulus information for tactile softness sensation of materials. *Cogn. Neurodyn.* **7**, 441–447 (2013).

Also, in discussion section, we added a sentence for noticing the future work of AiS-TSO for improvement as a bio-mimetic sensor in page 13 as follows: “This study shows a simple demonstration of filtering function by fabricating Fe-OFETs with the ferroelectric nanocomposite with the same concentration of BT NPs on a single substrate. Furthermore, AiS-TSO would be more practical if it is possible to fabricate an array of devices patterned with nanocomposites of different of BT NP concentrations by using additive printing process of improved nanocomposite solutions.”

7. The manuscript shows many schematic images of the device, but not one real photograph, SEM, TEM etc. image.

Response:

The comments on photographs or images are appreciated. To show the device image, we added the real photograph and cross-sectional TEM image of AiS-TSO on **Supplementary Figure 11**.

Supplementary Figure 11. Image of AiS-TSO. a, photograph of AiS-TSO with numbering the pixel. **b**, cross-sectional TEM (transmission electron microscopy) image of Fe-OFET in channel area prepared from the device structure by FIB (focused ion beam).

We added the sentence for indicating these images in manuscript (page8) as follows: “**The photograph of AiS-TSO and TEM image of cross-sectional view of Fe-OFET are shown in Supplementary Figure 11, and image of experimental setting for touch measurement is shown in Supplementary Figure 12.**”

8. It would be interesting to see the effect of moisture on the pressing finger. Is there any effect of the high permittivity of water on the function of the tactile sensor?

It seems that the response of the sensor to constant pressure is variable. How could this be fixed?

Response:

The comment related with environmental condition about humidity is appreciated. To confirm the effect of humidity on the device performance, we checked the touch response with varying the humidity of environment. Using a humidifier, we made the humid condition around from 60 RH % to 35 RH % in a box and checked the response to touch. The response was decreased when the humidity is increased, because the polyimide film is hydrophilic and as a result water adsorption can occur. In this case, the water adsorption on the PI surface increases surface charges so that discharges the surface. (Supplementary Information refs. 31 and 32) Therefore, adsorbed water layer prevents effective charge transfer between triboelectric layer (PI film) and skin decreasing the response of AiS-TSO. The experimental results were added in the revised supporting information as **Supplementary Figure 19** and references 31 and 32 in Supplementary Information were newly added.

Supplementary Figure 19. Measurement of AiS-TSO varying humidity condition.

Here we touched the AiS-TSO varying the humidity condition with around 1 kPa pressure. As shown in the **Supplementary Figure 19**, the response was decreasing with humidity increasing. This phenomenon can be explained the humidity effect on triboelectricity. Since the polyimide film is hydrophilic, so water absorption is high when it is in the high humidity condition increasing surface conductivity from

water layer³¹. The higher surface conductivity discharges the surface decreasing the effective triboelectric charges transfer between polyimide film and skin³².

Related references 31 and 32 were added in Supplementary Informaton.

31. Nguyen, V., Zhu, R. & Yang, R. Environmental effects on nanogenerators. *Nano Energy* **14**, 49-61 (2014). doi:10.1016/j.nanoen.2014.11.049
32. Nguyen, V. & Yang, R. Effect of humidity and pressure on the triboelectric nanogenerator. *Nano Energy* **2**, 604–608 (2013)

Also in manuscript to notice this result, we added the sentence in page 10: “By the way, since the triboelectric effect is highly influenced by humidity^{55,56}, we investigated the PSC response of AiS-Tso by finger touch in environments with different humidity levels. The results indicated that the I_{PSC} value was decreased with the humidity level increased (**Supplementary Figure 19**).”

Also, we added the references in manuscript 55 and 56 as follows:

55. Nguyen, V., Zhu, R. & Yang, R. Environmental effects on nanogenerators. *Nano Energy* **14**, 49-61 (2014). doi:10.1016/j.nanoen.2014.11.049
56. Nguyen, V. & Yang, R. Effect of humidity and pressure on the triboelectric nanogenerator. *Nano Energy* **2**, 604–608 (2013)

9. *The force measurement seems very basic using a hand force gauge. How did the authors ensure a constant force over time or a repetitive force of equal amplitude in case of varying forces?*

Response:

We appreciate the comment. To show how we applied similar and reliable force to the device, we added the image and video of measurement conditions. We tried to apply the intended force while checking the force gauge value. The picture is added on **Supplementary Figure 12**.

Supplementary Figure 12. Measurement setting for responses of AiS-TSO to touch. Measurement environment with force gauge and PCB board connected with the device.

We added the sentence to indicate this image in manuscript (page8) as follows: “**The photograph of AiS-TSO and TEM image of cross-sectional view of Fe-OFET are shown in Supplementary Figure 11, and image of experimental setting for touch measurement is shown in Supplementary Figure 12.**”

Reviewers' comments:

Reviewer #1 (Remarks to the Author):

The revised version of the author has clearly explained and basically satisfied the questions I asked.

Reviewer #2 (Remarks to the Author):

The reviewer highly appreciates the efforts that were undertaken and the additional measurements that were provided by the authors. Nevertheless, the authors cannot convincingly explain the nature of the observed triboelectric-capacitive coupling effect. They tend to relate that to ferroelectricity which obviously is not the cause due to several reasons (2.-5.)

1. The ambiguity in terms of whether there is a triboelectric-capacitive effect or mainly a pyroelectric or piezoelectric effect cannot be answered in a satisfying and unambiguous way by the provided experiments: In SI Fig 15c a finger touch with 1kPa load is compared to compressive or tensile bending (radius 2mm) to claim the minor contribution of piezoelectricity. That's like comparing apples to pears since transversal and longitudinal strain are quite different. It would have been more convincing to test the influence of a 1 kPa loading with an (insulating) PI stamp to a 1 kPa loading with a (conductive) finger.

2. Nevertheless, the reviewer also tends to believe that there is only minor contribution from Pyro/Piezoelectricity, especially since the authors write in their rebuttal that no polling/poling (no dipole alignment) was done before measurements. But, as I already stated in my first review, the authors talk about a natural poling occurring during manufacturing (Page 12, Methods section of original manuscript). So are the samples poled or not?

3. The polarization values that can be deduced from SI Fig. 7 for pure PVDF (3 mC/m²) are more than one order of magnitude smaller than published state of the art values, even the larger values for the composite (23 mC/m²) are a factor of 3-4 smaller than what is typical for pure PVDF-MFM structures (~ 80-100 mC/m²). Why this difference? Measurement method? Device structure?

4. It is not clear why a rather small voltage $V_{rec} = -10V$ (far below the coercive voltage for the composite layer of 0.6 μm (~23V)) should induce dipole alignment or even dipole switching as shown schematically in Fig.2b. To make that clear: dipole switching in ferroelectrics occurs at fields above the coercive field. In order to understand that, the authors should have had measured hysteresis in the same voltage range which they used for the characterization of the FeOFET in Fig.2. - Is there any ferroelectric hysteresis loop observable at voltage sweeps between +/-20V, +/-10V, +/-7V, ... like e.g. in SI Fig.7 for +/-45V? If not the observed effects have nothing to do with ferroelectric polarization switching and nothing to do with a remnant polarization that e.g. causes the current relaxation/retention times and long term plasticity.

It is rather probable that the high dielectric constant of the composite, which I assume to be far above 13 (the permittivity of pure PVDF-TrFE), and thus the high polarizability of the dielectric and not its ferroelectricity is responsible for all the observed effects. High-k materials are very prone to (triboelectric) charging and discharging/relaxation effects and thus capacitive coupling could work very efficient.

5. In this context, it also has to be emphasized that an OFET with a low-k PVP gate dielectric cannot be compared with an OFET with high-k dielectric like PVDF-TRFE/BT. And, as the authors state by themselves, the higher current level in the composite material (as compared to e.g. pure PVDF or PVP, why not indicating clearly the dielectric constant of the composite?) is due to its higher dielectric constant (Page 7 in the original manuscript). This is reasonable, because a higher gate dielectric constant naturally results in a lower threshold voltage and thus higher current levels.

6. Sentences like "Summing up the phenomena presented in Fig. 2, the state change of permanent polarization controlling the synaptic properties was dependent on the nature of the applied V_{rec} . The larger, longer, and more repetitive was the applied V_{rec} , the larger was the number of permanent dipoles switched above the coercive field and the greater is the permanent polarization generated, resulting in non-volatile memorization" are simply not true. Or "When V_{rec} was

removed, permanent polarization in the same direction as with V_{rec} was generated". This cannot happen with a V_{rec} (like -10V) that is too small to produce hysteresis, for fields much smaller than the coercive field no hysteresis loop opens up and no remnant polarization can be induced.

7. To sum up, the authors did not show the full hysteresis behaviour of the nanocomposite dielectric, especially not for the interesting voltage range (e.g. +/- 10V) neither did they provide values for the dielectric constant of this material. From the hysteresis curves provided (SI Fig.7, it can be assumed that ferroelectricity plays no role in the proposed AiI-TSO device, since most probably a hysteresis loop will not open up at $V_{rec} = +/- 10V$ (the typical V_{rec} value that was used for characterization) and no remnant polarization will develop. So the material then behaves rather like an electret or a paraelectric and the observed effects seem to be rather due to the high dielectric constant of the material.

Reviewer #3 (Remarks to the Author):

The authors have improved the manuscript. The writing is clearer, helping to understand the work. The comments of the reviewers have been addressed to some extent. Device images provided in supplementary fig. 11 are very interesting. The cross-section image shows an extremely inhomogeneous particle distribution and agglomeration. Large distances between NPs and channel can be observed, raising the question as to how they can affect the channel conductivity. The concept should be explained and confirmed by a theoretical model of the device (for instance FEM).

RESPONSE TO REFEREES (NCOMMS-19-17002B)

Reviewer #2

The reviewer highly appreciates the efforts that were undertaken and the additional measurements that were provided by the authors. Nevertheless, the authors cannot convincingly explain the nature of the observed triboelectric-capacitive coupling effect. They tend to relate that to ferroelectricity which obviously is not the cause due to several reasons (2.-5.)

1. The ambiguity in terms of whether there is a triboelectric-capacitive effect or mainly a pyroelectric or piezoelectric effect cannot be answered in a satisfying and unambiguous way by the provided experiments: In SI Fig 15c a finger touch with 1kPa load is compared to compressive or tensile bending (radius 2mm) to claim the minor contribution of piezoelectricity. That's like comparing apples to pears since transversal and longitudinal strain are quite different. It would have been more convincing to test the influence of a 1 kPa loading with an (insulating) PI stamp to a 1 kPa loading with a (conductive) finger.

Response:

The comments on the ambiguous experiments for explaining about triboelectric-capacitive effect are appreciated. We partly agree that the bending experiments in the **Supplementary Figure 17c** (changed from Supplementary Figure 15c) did not fully explain the triboelectric-capacitive coupling effect as the main mechanism of AiS-TSO. The reason we conducted the bending experiment in **Supplementary Figure 17c**, we touched the device loaded on a support fixture (**Supplementary Figure 14**) which has holes where the devices are located. The method was employed to prevent the support fixture from contacting the front of the device and to minimize the piezoelectric effect caused by the vertical force. So, we thought that if there is piezoelectric effect, it could be originated from tensile or compressive strain caused by bending of the device. Therefore, we checked the effect of tensile or compressive strain on the device response. However, we also thought that the bending experiment is not enough to prove that the main mechanism of AiS-TSO is

triboelectric-capacitive coupling effect. Therefore, as the reviewer suggested, vertical forcing test with PI (polyimide) can give the information on piezoelectric effect. In fact, we already conducted the experiment of touching by PI film with $\approx 1\text{kPa}$ force and showed the results in **Supplementary Figure 15h** when we revised the paper in the 1st revision. As shown in the graph (**Supplementary Figure 15h**), there was a very small response and negligible change of PSC (post synaptic current) with PI touching compared with touching by other materials.

Supplementary Figure 15. PSC dependence on touched materials. PSC when the AiS-TSO was touched with different materials (**a**, bare hand, **b**, gloved hand, **c**, aluminum foil, **d**, stainless steel foil, **e**, label tape paper, **f**, polyethylene naphthalate (PEN) film, **g**, polyvinyl chloride (PVC) film, **h**, polyimide film with a force of $\approx 1\text{ kPa}$).

To clearly explain this phenomenon, we added the explanation below the **Supplementary Figure 15**: “Furthermore, when we touched the device by polyimide film, the device response and change of PSC were very small because the triboelectric effect was the smallest. This phenomenon indicates that triboelectric-capacitive coupling is the dominant mechanism in AiS-TSO.”

2. Nevertheless, the reviewer also tends to believe that there is only minor contribution from Pyro/Piezoelectricity, especially since the authors write in their rebuttal that no poling/poling (no dipole alignment) was done before measurements. But, as I already stated in my first review, the authors talk about a natural poling occurring during manufacturing (Page 12, Methods section of original manuscript). So are the samples poled or not?

Response:

We did not carry out any intentional poling process to our device. To prove this, we conducted the additional experiment on the effect of device poling by biasing the gate electrode up to -30V for 30 min. After poling process, we compared the output characteristics of the poled and unpoled devices. As shown in **Supplementary Figure 9**, after poling process there is no current saturation region in the output characteristics of the poled device compared to the unpoled device (**Supplementary Figure 1b**). No saturation in the output characteristics (**Supplementary Figure 9a**) are attributed to increased accumulated holes by internal electric field from the poled gate insulator. Comparing of output characteristic of poled and unpoled device clearly indicate the polarization on the poled device, the unpoled device follows the typical output characteristics of FET with normal gate insulator even though it showed increased off-current and on-current levels compared to the device with PVP gate dielectric. (**Supplementary Figure 3**) We described that the as-fabricated device is naturally poled based on higher off-current compared to the device with no ferroelectric gate dielectric (i.e with PVP). But the comparison of poled and unpoled device characteristics clearly show that the degree of natural poling in the as-fabricated device with ferroelectric gate is not significant compared to the poled device. Relatively high off-current may be attributed to rougher surface

compared to that of PVP. The description on natural poling was a misjudgment without experiment proof. We are sorry about causing the confusion. We erased the sentence on natural poling during fabrication.

Since we do not use the mechanism of piezoelectric effect to induce the drain current modulation of the AiS-TSO, we do not need to generate the saturated remnant polarization in the ferroelectric gate insulator before driving the device. In our device, partial polarization in gate insulating layer is used for generating synaptic weight. The partial polarization is also related to mechanism of previously reported memory devices using intermediate states in minor loop of ferroelectric materials (Supplementary Information ref.9,18,28-32). As ferroelectric memory devices do not require poling process before driving them (ref. 31-39 in manuscript), we used the device without poling process. Rather, we could observe that synaptic properties were much different for the poled and unpoled device. As shown in **Supplementary Figure 9c and 9d**, when we applied the 100 pulses (pulse width of 0.1 s, amplitude of -10 V or 10V) on gate electrode of the poled and unpoled device, behaviors in drain current and synaptic weight were much different. Synaptic property could not be obtained by V_{rec} (in the same direction of internal field) in the poled device since the dipoles are already switched (**Supplementary Figure 9c**), which strongly indicate the effect of poling process. When the V_{rec} pulses are applied in the opposite direction to the internal field, partial switching of dipoles occurred resulting in a smaller synaptic weight than that of unpoled device (**Supplementary Figure 9d**). Those results imply that the status of switched dipoles affect the synaptic properties.

We added **Supplementary Figure 9** to describe comparison the characteristics of the poled and unpoled devices by observing the PSC change and output curve according to the voltage pulse.

Supplementary Figure 9. Characteristics of Fe-OFET after poling process and comparison of PSC values before and after device poling. **a**, Output characteristics of poled Fe-OFET (using BT NPs 20 wt%/P(VDF-TrFE) composite as gate insulator). The poling was carried out by applying the negative gate bias of -30 V for 30 min. **b**, Dipole switching after poling process by applying the field between gate and drain electrode and formation of internal field. PSC change ratios of unpoled (red) and poled (blue) Fe-OFET by applying 100 pulses of V_{rec} with the amplitude of **c**, -10 V and **d**, 10 V (pulse width of 0.5 s).

Here in **Supplementary Figure 9**, we compared PSC values for the device with poled ferroelectric gate dielectric layer. We conducted poling process by applying the bias of -30 V between gate electrode and drain electrode for 30 min. The channel layer acts as a poling electrode. As shown in **Supplementary Figure 9a**, output characteristics showed increased drain current without saturation, which is different from

saturation behavior of unpoled device. No saturation in the poled device is attributed to internal field generated by remnant polarization of gate insulating layer. This result corresponds to our previous investigation about poled Fe-OFET.^{26,27} The generated internal field in the poled device acts as negative bias, which enhances accumulation of holes in p-type organic semiconductor channel (**Supplementary Figure 9b**). PSC change and synaptic properties with V_{rec} pulsing were much different for the unpoled and poled device. During pulsing of -10 V V_{rec} is applied to the poled device, the change of PSC is negligibly small since the dipoles in the ferroelectric layer are already aligned (**Supplementary Figure 9c**). So, the dipoles are difficult to be switched further in the same direction with negative V_{rec} pulsing since they are already fully switched by gate biasing of -30 V. Therefore, the synaptic weight of the poled device under negative V_{rec} pulsing is negligible because the polarization was already saturated by poling process. During positive V_{rec} pulsing, the change of PSC in poled device is much smaller than that in unpoled device. The positive V_{rec} pulsing will try to rotate dipoles in the opposite direction to the poled direction of the poled ferroelectric layer resulting in a slight decrease in the PSC because partial switching of dipoles can be more difficult by applying the field opposite to aligned direction of dipoles compared to switching of randomly oriented dipoles (**Supplementary Figure 9d**). Under positive V_{rec} pulsing of the poled device, therefore, a smaller synaptic weight value than that of the unpoled device was also observed. In our approach, partial polarization in ferroelectric gate insulating layer in the unpoled device is used for generating synaptic weight by applying negative or positive V_{rec} pulses. The partial polarization behavior is closely related to mechanism of ferroelectric memory devices.^{9,28-32} Therefore, we conclude that the poling process was disadvantageous in generating synaptic weights with potentiation and depression. In our work, the unpoled devices were used for all other measurements to utilize the change in partial polarization as synaptic plasticity.

Related to description of **Supplementary Figure 9**, we added the references in supplementary information as follows:

9. Kim, E. J., Kim, K. A. & Yoon, S. M. Investigation of the ferroelectric switching behavior of P(VDF-TrFE)-PMMA blended films for synaptic device applications. *J. Phys. D. Appl. Phys.* **49**, 075105 (2016).
26. Tien, N. T. *et al.* A flexible bimodal sensor array for simultaneous sensing of pressure and

temperature. *Adv. Mater.* **26**, 796–804 (2014).

27. Tien, N. T., Trung, T. Q., Seoul, Y. G., Kim, D. Il & Lee, N. Physically responsive field-effect Transistors with Giant Electromechanical Coupling Induced by Nanocomposite Gate Dielectrics. *ACS Nano* **5**, 7069–7076 (2011).
28. Zeng, B. *et al.* 2-bit/cell operation of Hf_{0.5}Zr_{0.5}O₂ based fefet memory devices for NAND applications. *IEEE J. Electron Devices Soc.* **7**, 551–556 (2019).
29. Tomas, J., Bellaiche, L. & Bibes, M. Learning through ferroelectric domain dynamics in solid-state synapses. *Nat. Commun.* **8**, 14736 (2017).
30. Kim, M. K. & Lee, J. S. Ferroelectric analog synaptic transistors. *Nano Lett.* **19**, 2044–2050 (2019).
31. Jerry, M., Dutta, S., Kazemi, A., Ni, K. & Zhang, J. A ferroelectric field effect transistor based synaptic weight cell. *J. Phys. D. Appl. Phys.* **51**, 433001 (2018).
32. Duiker, H. M. *et al.* Fatigue and switching in ferroelectric memories : Theory and experiment fatigue and switching in ferroelectric memories *J. Appl. Phys.* **68**, 5783 (1990).

To make sure that we did not conduct the poling process on the device to use partial polarization mechanism efficiently for generating synaptic weight, we added the sentences in page 7 as follows: “Since we used partial polarization of ferroelectric gate insulating layer, we did not conduct poling process. In case of generating SW by using the partial polarization, poling process on the device which makes the saturated polarization switching resulting in formation the internal field in gate insulating layer was disadvantageous for generation of SW with both potentiation and depression. (Supplementary Figure 9).”

Also, we have provided misunderstanding by mentioning that “An additional poling process was not applied because poling naturally occurred during the device fabrication process.” in page 15 in Methods section, we revised the sentence as follows: “We did not conduct poling process for gate insulator.”

3. The polarization values that can be deduced from SI Fig. 7 for pure PVDF (3 mC/m²) are more than one order of magnitude smaller than published state of the art values, even the larger values for the composite (23 mC/m²) are a factor of 3-4 smaller than what is typical for pure PVDF-MFM structures (~ 80-100 mC/m²). Why this difference? Measurement method? Device structure?

Response:

We appreciate the comment on the polarization values of the device. We realized that we made a mistake in calculating the polarization values depending on the electrode area of the device. In the metal-insulator-metal (MIM) device fabrication process, we used a shadow mask which has various sizes of square shape for deposition of top electrode. When we measured the P-E curve, we input the area size of 1 cm² in the software of measurement system, and then, we multiplied the factors for matching with the actual electrode area to plot the polarization curves. In calculation process, since the devices with various electrode areas are configured on the same substrate, we were confused about the exact size and made a mistake in the calculations, resulting in abnormal results. We apologize for this confusion.

Therefore, we measured P-E data again, and the correct device size was entered into the software of measurement system from the beginning to eliminate the need for proportional calculations and to make more accurate P-E measurement. The structure of device was MIM and the thickness of the insulating layer was 600 nm, and the thickness of top and bottom electrodes were 65 nm. Also, we checked the polarization curve with the area varied in **Supplementary Figure 7f** and we used a mask with the same electrode area of 0.0025 cm² to minimize the size dependency on polarization for all other P-E curve measurements (**Supplementary Figure 7a-e**). We added re-measured P-E curves in new **Supplementary Figure 7** (changed from Supporting Figure 7 in the previous version). The remnant polarization for the structure with P(VDF-TrFE) (~ 40 mC/m²) was smaller than the value mentioned by the reviewer (~ 80-100 mC/m²). However, the obtained value is similar to other values reported by other groups (Supplementary Information ref.8-12). We presume that the difference in polarization values from the value mentioned by the reviewer might be due to the difference in the thickness of the insulator, the type of metal and the voltage applied (Supplementary Information ref.8-12). We could confirm that the coercive field of P(VDF-TrFE) was larger than that of BT NPs/P(VDF-TrFE) composite. Also, remnant polarization of BT NPs/P(VDF-TrFE) was larger than that of P(VDF-TrFE), as we expected.

The P-E curves were corrected in **Supplementary Figure 7a**, and we added the top view image of MIM structure device to clarify the size of the device (**Supplementary Figure 7h**). Also, the explanation of the observed phenomenon was edited below the figures.

Supplementary Figure 7. Fundamental characteristics of polarization-electric field (P-E) curve of ferroelectric films. **a**, P-E curves of BT NP(20 wt%)/P(VDF-TrFE) and P(VDF-TrFE) thin films with the thickness of 600 nm. **b**, P-E curves of BT NP(20 wt%)/P(VDF-TrFE) and P(VDF-TrFE) thin films by applying -10 V to 10 V as driving voltage. **c**, P-E curves of BT NP(20 wt%)/P(VDF-TrFE) nanocomposite thin film with the applied voltage varied. The sub-loops of P-E at the applied voltage range of -5 V ~ 5 V to -60 ~ 60 V were also included. **d**, The sub-loops of P-E curve of BT NP(20 wt%)/P(VDF-TrFE) nanocomposite thin film at the applied voltage range of -1 V ~ 1 V to -10 V ~ 10 V. P-E curve of BT NP(20 wt%)/P(VDF-TrFE) nanocomposite thin film at the applied voltage range of -10 V to 10 V with the **e**, frequency varied of applied voltage **f**, and electrode area of devices. **g**, Schematic of device structure for

measurement of P-E curve as MIM (metal-insulator-metal) structure. **h**, Optical image of MIM device at top view.

To confirm the effect of BT NPs in the nanocomposite film on polarization behaviors, we fabricated the metal (Pt)-ferroelectric-metal (Al) (MFM) structures with the ferroelectric layers of BT NP(20 wt%)/P(VDF-TrFE) nanocomposite and P(VDF-TrFE) on SiO₂/Si wafer and measured polarization-electric field (P-E) curves by applying the voltage from -80 V to 80 V (**Supplementary Figure 7a**). The smaller coercive field (E_c) of 48.83 ($-E_c = -45.83$) MV/m for BT NP(20 wt%)/P(VDF-TrFE) nanocomposite film was obtained compared to that ($E_c = 88.2$ MV/m, $-E_c = -84.6$ MV/m) for pure P(VDF-TrFE) film. Also, the nanocomposite film has much larger remnant polarization (P_r) ($6.1 \mu\text{C}/\text{cm}^2$) than that of pure P(VDF-TrFE) ($4 \mu\text{C}/\text{cm}^2$). These results demonstrate that generation of larger P_r in BT NP (20 wt%)/P(VDF-TrFE) was observed compared to pure P(VDF-TrFE), which means that synaptic behaviors of the Fe-OFET can be tuned by varying the concentration of BT NPs in the nanocomposite. There would be some differences in P_r compared to the values from the previous reports about P(VDF-TrFE) thin films⁸⁻¹¹. This could occur from different thickness of film, electrode area of device or measurement condition such as driving voltage.^{8,10,11} However, the coercive field value was not much different from those in the previous reports and the tendency of increasing in polarization was observed when the BT NPs are included in nanocomposite, similarly to the results reported from the previous reports⁸⁻¹². Also, when the range of applied voltage was varied to a smaller range, there was observed partial polarization in minor loop. We could demonstrate that usage of smaller range of voltage such as -10 V to generate SW by controlling the partial polarization with pulse rate, number and duration time varied. Therefore, polarization switching can be controlled according to the range of applied voltage¹³⁻¹⁷, which implies that SW in Fe-OFET will depend on amplitude, duration time, rate and number of the V_{rec} pulses similar to other Fe-RAM devices using minor loop of ferroelectric materials.^{9,18-22} Of course, the device has smaller retention time for polarization of minor loop¹⁸⁻²¹. However, the retention time can be controlled by pulse duration time, number or frequency for generating SW and controlling the STP and LTP behaviors, as shown in **Fig. 2**.

Related to description in **Supplementary Figure 7**, we added some references (8,10-12,18-22) in

Supplementary Information as follows:

8. Fujisaki, S., Ishiwara, H. & Fujisaki, Y. Low-voltage operation of ferroelectric poly(vinylidene fluoridetrifluoroethylene) copolymer capacitors and metal-ferroelectric- insulator-semiconductor diodes. *Appl. Phys. Lett.* **90**, 162901 (2007).
10. Hu, W.J. *et al.* Universal ferroelectric switching dynamics of vinylidene fluoride-trifluoroethylene copolymer films. *Sci. Rep.* **4**, 4772 (2014).
11. Yoon, S. *et al.* Nonvolatile memory thin-film transistors using an organic ferroelectric gate insulator and an oxide semiconducting. *Semicond. Sci. Technol.* **26**, 4869-4874 (2011).
12. Valiyaneerilakkal, U. & Varghese, S. Poly (vinylidene fluoride-trifluoroethylene) / barium titanate nanocomposite for ferroelectric nonvolatile memory devices poly(vinylidene fluoride-trifluoroethylene)/barium titanate nanocomposite for ferroelectric nonvolatile. *AIP. Adv.* **3**, 11–16 (2014).
18. Oh, S., Kim, T., Kwak, M., Song, J. & Woo, J. HfZrO_x-Based ferroelectric synapse device with 32 levels of conductance states for neuromorphic applications. *IEEE Electron Device Lett.* **38**, 732–735 (2017).
19. Lee, D. *et al.* Multilevel data storage memory using deterministic polarization control. *Adv. Mater.* **24**, 402–406 (2012).
20. Zhao, D. *et al.* Retention of intermediate polarization states in ferroelectric materials enabling memories for multi-bit data storage. *Appl. Phys. Lett.* **108**, 232907, (2016).
21. Lee, K. *et al.* Stable subloop behavior in ferroelectric Si-Doped HfO₂. *ACS Appl. Mater. Interfaces* **11**, 38929–38936 (2019).
22. Kyunys, B., Lurchuk, V., Meny, C., Majjad, H., & Doudin, B. Sub-coercive and multi-level ferroelastic remnant states with resistive readout. *Appl. Phys. Lett.* **104**, 232905 (2014).

Also, according to updated results, we edited the coercive field of BT NP(20wt %)/P(VDF-TrFE) and P(VDF-TrFE) film in manuscript as follows (page 8): “We found that the BT NP(20 wt%)/P(VDF-TrFE) nanocomposite has a smaller coercive field (48.83 MV/m) than that of P(VDF-TrFE) only (88.2 MV/m), which implies easier polarization switching in the nanocomposite than in P(VDF-TrFE).”

4. *It is not clear why a rather small voltage $V_{rec}=-10V$ (far below the coercive voltage for the composite layer of $0.6\ \mu m$ ($\sim 23V$)) should induce dipole alignment or even dipole switching as shown schematically in Fig.2b. To make that clear: dipole switching in ferroelectrics occurs at fields above the coercive field. In order to understand that, the authors should have had measured hysteresis in the same voltage range which they used for the characterization of the FeOFET in Fig.2. - Is there any ferroelectric hysteresis loop*

observable at voltage sweeps between +/-20V, +/-10V, +/-7V, ... like e.g. in SI Fig.7 for +/-45V? If not the observed effects have nothing to do with ferroelectric polarization switching and nothing to do with a remnant polarization that e.g. causes the current relaxation/retention times and long term plasticity.

Response:

We appreciate the comments about polarization in ferroelectric materials related to coercive field. As you mentioned, saturated dipole switching could be achieved only above coercive field and remnant polarization in that case remains long and stable. Therefore, many researchers have tried to make ferroelectric memory devices which have remnant polarization with long retention time by applying write voltage that can make saturated dipole switching (ref. 32 in manuscript). However, in our device, we used partial polarization switching formatted by applying repetitive (not single) pulse biasing below the coercive field to generate and control the synaptic weight. Partial polarization switching before formation of saturated dipole switching by applying electrical field below the coercive field can be explained in minor loops of P-E curve. Although the polarization value formed in the minor loop region is small and the retention time is not long, we conducted most of the experiments on synaptic weight formation and regulation by applying a voltage of -10 V, the magnitude of the voltage in the minor loop region. Therefore, we totally agree with reviewer's comment that we should check the P-E curve at voltage sweeps between +/-20V, +/-10V, +/-7V and so on. Accepting the comments, we conducted further P-E curve measurements and added the results in **Supplementary Figure 7**. As shown in **Supplementary Figure 7b**, we conducted measurement of P-E curves of both BT NP(20 wt%)/P(VDF-TrFE) and P(VDF-TrFE) at voltage switching +/-10V and confirmed the sub-loop polarization in BT NP(20 wt%)/P(VDF-TrFE) ($\sim 0.02 \mu\text{C}/\text{cm}^2$) was larger than that of P(VDF-TrFE) ($\sim 0.006 \mu\text{C}/\text{cm}^2$). Although the polarization formed in minor loop is very small, we could check the tendency of varied the range of sweep voltage in minor loops which were similar to other reports (ref.33-39 in manuscript) on minor loops and partial polarization switching of ferroelectric materials (**Supplementary Figure 7c**). We further checked the hysteresis in below +/-10V of switching voltage and partial polarization switching in minor loops increases when increasing the voltage (**Supplementary Figure 7d**). We used the

partial polarization switching in minor loops to generate and control synaptic weight in our device by controlling the duration, number and frequency of applied voltage related with ferroelectric switching dynamics (ref. 30,32,38 in manuscript). Therefore, we agree that the retention time of polarization in our device is not so long since it is formed in minor loop. However, we believed that we could control it and usefully applied to demonstrate the synaptic property. The reason we use the voltage range in minor loop can be described in two ways. First of all, if we use the bias pulse above coercive voltage to earn the partial polarization switching and synaptic weight, we expected that the window of synaptic weight and PSC increasing ratio would be decreased. To demonstrate this and help the reader understand, we measured the PSC with -30V of V_{rec} pulsing (100 times, pulse width of 0.5 s) on gate electrode (coercive voltage was around -27.5 V).

Also, we checked the retention time and compared it with -10 V V_{rec} pulsing and added the results in **Supplementary Figure 8**. As shown in **Supplementary Figure 8**, when we applied 100 V_{rec} pulses of -30 V, the linearity of the increasing ratio of the PSC peak value was saturated very quickly and hardly increased at the end of the pulsing. On the other hand, when we applied 100 V_{rec} pulses of -10 V, the linearity of increasing ratio in PSC was more stable and not fully saturated until the end of the pulse. Of course, the retention time of polarization from partial polarization switching was short in case of -10 V V_{rec} pulsing. Compared to the retention time which takes for the increased PSC ratio to fall below 15% immediately after applying the voltage, it was about 68 min for -10 V of V_{rec} and 1814 min for -30 V of V_{rec} . Clearly, for pulsing of V_{rec} of -30 V the inherent PSC increasing was maintained for much longer period of time, and the maximum inherent PSC increasing rate was also about 2 times larger. However, as we previously explained, the linearity of the increasing rate decreases and the instability of the current increases. In addition, one of the major problems of other ferroelectric-based memory devices has been reported to be some problems such as ferroelectric fatigue and device breakdown when the write / erase processes are repeated with such a large voltage (ref. number 32-39 in manuscript). Therefore, we determined that it would not be necessary to create synaptic weight by applying a voltage above the coercive voltage. This is because synaptic weight can be formed by controlling the time, number, and speed of the pulse sufficiently, and even at -10 V of V_{rec} , it could be proven as a device that mimics the memory function of the human

body's sensory organs. In terms of synaptic plasticity, the long-term potentiation is defined in wide ranges from minutes to years (ref. number 27,40 and 61 in manuscript). Also, in this study, we conducted a research on the fact that we can have memory function in addition to sensory function even for several hours, rather than implementing semi-permanent memory function like decades. Secondly, we choose -10 V of V_{rec} because we cannot induce above coercive voltage from touching the polyimide (**Supplementary Figure 19**). Since the triboelectric-capacitance coupling effect between PI and skin is the main mechanism of AiS-TSO, so we had to choose a suitable voltage which can be generated by triboelectric effect between PI and skin to prove the synaptic characteristics in AiS-TSO through Fe-OFET.

To sum up, we checked the polarization in minor loops to demonstrate that generation of synaptic weight is originated from partial polarization (**Supplementary Figure 7**). Even though partial polarization is small, it could be controlled with time, frequency and number of pulses following switching dynamic in ferroelectric material (ref. 30-39 in manuscript). Also, to clarify the reason we chose the -10 V of V_{rec} which is voltage in minor loop, we conducted extra experiment to compare the PSC with pulsing -30 V of V_{rec} which is above the coercive voltage. When we apply the -30 V of V_{rec} , even though the retention time was longer, the linearity of increasing ratio of PSC was poor and unstable. Therefore, we could support the reason that we used -10 V of V_{rec} to demonstrate synaptic weight since it is enough to demonstrate the memory function added on detection function mimicking our sensory organ's function from synapse like connection between Merkel cell and afferent neuron. We added those explanation with **Supplementary Figure 8** as follows:

Supplementary Figure 8. Comparison PSC and retention time with applying pulse of -10 V and -30 V of V_{rec} . **a**, PSC change when applying the 100 pulses of V_{rec} (amplitude of -10 V, pulse width of 0.5 s). **b**, PSC change when applying the 100 pulses of V_{rec} (amplitude of -30 V, pulse width of 0.5 s). The retention time of PSC and PSC change after pulsing of **c**, -10 V and **d**, -30 V of V_{rec} .

Here we checked the PSC values when applying -30 V of V_{rec} which is above the coercive voltage of gate insulating layer in Fe-OFET. Comparing with the results obtained with -10 V of V_{rec} , the results show the fast saturation in PSC but poor characteristics in terms of linearity of PSC increasing rate. On the other hand, in case of retention time, the time taken to drop below 15% of the maximum PSC change at -30 V was longer (~1814 min) than that at -10 V (~68 min). Also, the maximum PSC increasing rate was also about 2 times at -30 V of V_{rec} (~18) larger than that at -10 V of V_{rec} (~11). This is related to remnant polarization formed by applying bias pulses above coercive field and, therefore, much longer retention time

was obtained compared to that by applying bias pulses in minor loops of P-E curve. But as we mentioned before, we can generate and adjust SW even with -10 V of V_{rec} pulses. Therefore, we judged that it was not necessary to apply a voltage above the coercive field, which would decrease the linearity of the PSC increasing rate and the SW window, and cause the fatigue in ferroelectric material resulting in device breakdown. Also, we used the triboelectric-capacitive coupling effect by touching the polyimide substrate to generating of SW on AiS-TSO, and the measurement of synaptic characteristics of Fe-OFET was conducted as electrical demonstration of AiS-TSO. We could not generate triboelectric voltage output above coercive field by touching. So, we determined that using -10 V magnitude of V_{rec} is appropriate to demonstrate the generation of the synaptic weight originated from ferroelectric dielectric layer. Furthermore, we focus on the fact that our sensory organs have synaptic-like functions that can be used for sensory memory before they are processed in the brain, which are not intended to implement semi-permanent memory. Therefore, we believe that using -10 V of V_{rec} in the minor loop region is enough to demonstrate synaptic function by mimicking the Merkel cell neurite complex. LTP in synapse is very broad²³⁻²⁵, ranging from minutes to decades, and we can control the STP (seconds) and LTP (hours) of this device by controlling the duration, number and frequency of stimuli.

Related to the description in **Supplementary Figure 8**, we added some references (23-25) as follows in Supplementary Information:

23. Zucker, R. S. & Regehr, W. G. Short-term synaptic plasticity. *Annu. Rev. Physiol.* **64**, 355–405 (2002).
24. Schall, J. D. Neural basis of deciding, choosing and acting. *Nature Rev. Neurosci.* **2**, 33-42 (2001).
25. Mcgaugh, J. L. Memory — a century of consolidation. *Science* **287**, 248-251 (2000).

Also, we revised the manuscript to clarify that we used the partial polarization of minor loop in ferroelectric materials and reason of we choose the ± 10 V(voltage range in minor loop region)for most demonstration of synaptic weight as follows in page 6: “**Generation of SW in the AiS-TSO with ferroelectric nanocomposite can be explained by partial polarization switching at low electric field, similarly to previous investigations on polarization switching dynamics of minor loops in ferroelectric materials³⁰⁻³⁵.** Further

investigation of ferroelectric polarization with measurement of P-E curves was conducted (**Supplementary Figure 7**). We investigated the partial polarization in minor loops of BT NP(20 wt%)/P(VDF-TrFE) thin film which can be used to generate SW in our device by controlling the amplitude, duration time, frequency and number of pulses of V_{rec} . We usually use the pulse amplitude of ± 10 V using minor loops of ferroelectric layer rather than the saturation loop since we could get synaptic properties which can be used for sensory memory^{8,23,24,40} by controlling the partial polarization switching in ferroelectric gate dielectric layer and so control the STP and LTP properties by varying the duration, number and frequency of pulses (**Fig 2**). Of course, when we compare the memory retention time taken to drop below 15% of the maximum PSC change of I_{PSC} , the retention time was much smaller (~ 68 min) with -10 V of V_{rec} pulses applied than that with the pulse amplitude of -30 V was applied (retention time ~ 1814 min) (**Supplementary Figure 8**). The results are consistent with the concerns about using minor-loop of ferroelectric materials addressed in the previous reports^{33,36,37}. However, high linearity of PSC was advantageous when we use the -10 V than -30 V as V_{rec} pulse amplitude as shown in **Supplementary Figure 8**. In addition, to demonstrate that the triboelectric-capacitance coupling effect between PI and skin contributes to the generation of SW in AiS-TSO, we had to choose a suitable bias pulse voltage in Fe-OFET similar to the potential obtained by triboelectric effect between PI and skin. For those reasons, we chose ± 10 V of V_{rec} pulse for demonstration of synaptic properties of Fe-OFET.”

In addition, related with this revised manuscript, we remove the references 31, 33, and 35 in the previous version of manuscript and added new references (33-39) with editing the references number as follows:

33. Jerry, M., Dutta, S., Kazemi, A., Ni, K. & Zhang, J. A ferroelectric field effect transistor based synaptic weight cell. *J. Phys. D. Appl. Phys.* **51**, 433001(2018).
34. Kim, M. & Lee, J. Ferroelectric analog synaptic transistors. *Nanolett.* **19**, 2044-2050 (2019).
35. Kyunys, B., Lurchuk, V., Meny, C., Majjad, H., & Doudin, B. Sub-coercive and multi-level ferroelastic remnant states with resistive readout. *Appl. Phys. Lett.* **104**, 232905 (2014).
36. Kim, E. J., Kim, K. A. & Yoon, S. M. Investigation of the ferroelectric switching behavior of P(VDF-TrFE)-PMMA blended films for synaptic device applications. *J. Phys. D. Appl. Phys.* **49**, 075105 (2016).
37. Zhao, D. *et al.* Retention of intermediate polarization states in ferroelectric materials enabling memories for multi-bit data storage. *Appl. Phys. Lett.* **108**, 232907, (2016).

38. Oh, S., Kim, T., Kwak, M., Song, J. & Woo, J. HfZrO_x-Based ferroelectric synapse device with 32 levels of conductance states for neuromorphic applications. *IEEE Electron Device Lett.* **38**, 732–735 (2017).
39. Yoon, S., Kim, E., Kim, Y. & Ishiwara, H. Adaptive-learning functions of ferroelectric field-effect transistors for synaptic device applications. *2017 International Symposium on Nonlinear Theory and Its Applications*, 314-317, (2017).

5. *It is rather probable that the high dielectric constant of the composite, which I assume to be far above 13 (the permittivity of pure PVDF-TrFE), and thus the high polarizability of the dielectric and not its ferroelectricity is responsible for all the observed effects. High-k materials are very prone to (triboelectric) charging and discharging/relaxation effects and thus capacitive coupling could work very efficient.*

Response:

We agree that BT NPs/P(VDF-TrFE) insulating layer has high polarizability as dielectric layer compared to PVP insulating layer. In our device, triboelectric effect is not directly caused by the ferroelectric layer since we do not touch the ferroelectric layer by finger but touch polyimide layer by finger. Dielectric constant of the touched triboelectric layer by finger (here it is polyimide) may affect the degree of triboelectric-capacitive coupling in which charges induced by triboelectric under finger touch are stored in the PI layer. Of course, high-k ferroelectric materials as gate insulators help to increase the drain current level in OFET. However, we would like to say that high dielectric constant gate insulator increases the PSC during pulsed or touched (on-state current) due to higher gate capacitance but not after finishing the stimuli (off-state current). In our device, the inherent change of conductivity in channel layer is induced from residual internal field from polarization in ferroelectric material coupled by triboelectric-capacitive coupling effect as we have shown by poling and partial polarization experiments.

And, if there is no ferroelectric effect on synaptic effect, one possible mechanism is the charge trapping in the gate dielectric or in the channel. For synaptic effect in the FET, residual internal field with a certain degree of retention after removal of stimuli should be generated in the gate dielectric or channel to cause change in drain current. So, we tried to find other reports on the effect of dielectric constant of gate

dielectric on synaptic properties of synaptic transistors based on the charge trapping mechanism, but we could not find the effect of dielectric constant of gate insulating layer on synaptic properties. Investigation of the effect of the gate dielectric constant on synaptic properties of charge trapping mechanism in synaptic transistors requires another set of experiment to figure out the mechanism. At least, we could not see the significant synaptic properties from the device with low-dielectric constant. And, we could not explain the synaptic properties of our device after finishing the applying voltage as the effect of higher dielectric constant. Furthermore, investigation of synaptic properties in poled devices (responses to the comment #2) indicated that partial polarization can change the synaptic behaviors under positive and negative V_{rec} pulses, which indicates the switching of dipoles under electrical pulsing is an important attribute in Fe-FET device.

6. In this context, it also has to be emphasized that an OFET with a low-k PVP gate dielectric cannot be compared with an OFET with high-k dielectric like PVDF-TRFE/BT. And, as the authors state by themselves, the higher current level in the composite material (as compared to e.g. pure PVDF or PVP, why not indicating clearly the dielectric constant of the composite?) is due to its higher dielectric constant (Page 7 in the original manuscript). This is reasonable, because a higher gate dielectric constant naturally results in a lower threshold voltage and thus higher current levels.

Response:

We appreciate the comment on effect of higher dielectric constant on synaptic weight. The BT NPs/P(VDF-TrFE) has higher dielectric constant than that of PVP. As shown in **Supplementary Figure 3**, we could see the effect of dielectric constant on hysteresis in transfer characteristics and PSC change of Fe-OFET with BT NPs/P(VDF-TrFE) and OFET with PVP gate dielectric layer. The purpose of our comparative analysis of OFETs using PVP and BT NP(20 wt%)/P(VDF-TrFE) was to prove the principle of synaptic weight generation by characterizing transistor devices using ferroelectric and non-ferroelectric materials.

Higher dielectric constant makes the drain current value larger in the transfer characteristics (**Supplementary Figure 3a**) and larger change in inherent PSC during V_{rec} pulsing due to larger gate

capacitance of high-k dielectric constant gate insulator (**Supplementary Figure 3b**) compared to PVP device. However, there are negligible hysteresis in the transfer curve of PVP device (**Supplementary Figure 3a**), which is related to charge trapping and, in turn, synaptic weight generation. These results indicate that there is negligible formation of residual internal field by charge trapping after finishing the pulsing, differently from Fe-FET with BT NPs/P(VDF-TrFE). We agree that there is difference in dielectric constants of two materials, but synaptic weight is originated from the residual internal electric field formed after terminating V_{rec} pulses (**Fig. 2b** in manuscript).

In this respect, we would like to explain the inherent change in the PSC after finishing V_{rec} pulsing is originated from formation of switched partial polarization, not from the high dielectric constant. As we explained in the response to the comment #5, if synaptic weight generation is not by ferroelectric effect, we may explain the effect by charge trapping mechanism in organic semiconductor (Supplementary Information refs. 6,7). However, this mechanism cannot fully explain the effect of poling as we described in the response to the comment #2. Therefore, this phenomenon may not be related to dielectric constant but is related to residual internal field by partial polarization. So, we would like to explain the intention of our experiment in **Supplementary Figure 3** by looking at the synaptic weight after pulsing finished and hysteresis in transfer curve. Due to the possibility of misunderstanding to readers, we edited the description in **Supplementary Figure 3** by adding the capacitance values of PVP and BT NP(20 wt%)/P(VDF-TrFE) which were measured by MIM structure to explain the reason of higher PSC during the pulsing.

Supplementary Figure 3. Comparison of characteristics OFET with ferroelectric and non-

ferroelectric gate dielectric. a, Transfer curves of OFET with non-ferroelectric PVP (polyvinylpyrrolidone) gate dielectric and Fe-OFET with the ferroelectric nanocomposite gate dielectric (BT NP(20 wt%)/P(VDF-TrFE)). **b**, Change of PSC (Δ PSC) in OFET with non-ferroelectric PVP gate dielectric and Fe-OFET with the ferroelectric nanocomposite gate dielectric (BT NP(20 wt%)/P(VDF-TrFE)) when the 1.42 Hz of pulses are applied on gate electrode (pulse width = 0.5 s, amplitude = -10 V).

To demonstrate that ferroelectric characteristics of gate dielectric layer in Fe-OFET mainly contribute to synaptic properties of the device, we also fabricated the OFET with non-ferroelectric PVP (polyvinylpyrrolidone) as a gate dielectric layer. The measured capacitance of BT NP(20wt%)/P(VDF-TrFE) was much higher (~ 21 nF/cm²) than PVP (~ 5.2 nF/cm²) in the MIM structure with the same insulator thicknesses as those in the FET structure, which results in the higher on-state current level in the transfer characteristics (**Supplementary Figure 3a**) in Fe-OFET with ferroelectric nanocomposite than that in OFET device with PVP device. The change of PSC (Δ PSC) of the device with PVP ($\sim 1 \times 10^{-11}$ A) during V_{rec} biasing was much smaller than that of the device with BT NP(20 wt%)/P(VDF-TrFE) ($\sim 1 \times 10^{-9}$ A) during V_{rec} biasing (**Supplementary Figure 3b**). However, Fe-OFET with ferroelectric gate dielectric material has larger hysteresis in the transfer curve (**Supplementary Figure 3a**) and larger change in PSC after finishing V_{rec} pulsing (**Supplementary Figure 3b**) compared to OFET with PVP gate insulator. These results are originated from internal field generated in partial polarization switching in ferroelectric material, which results in the generation of synaptic weight (SW) in the Fe-OFET. On the other hand, negligible hysteresis and residual Δ PSC after finishing V_{rec} pulsing in the OFET with non-ferroelectric PVP indicate that charge trapping does not significantly affect synaptic properties as much as ferroelectric effect event though charge trapping of organic semiconductors has been reported.^{6,7} On the other hand, it can be argued that synaptic property in our Fe-OFET device is mainly related to ferroelectricity of nanocomposite gate dielectric layer.

Also, we added the sentence to explain the equipment which we used for measurement of capacitance at Measurements section in manuscript as follows (page 15): “Capacitance measurement with MIM device was conducted using a semiconductor parameter analyzer (Agilent, B1500).”

7. Sentences like "Summing up the phenomena presented in Fig. 2, the state change of permanent polarization controlling the synaptic properties was dependent on the nature of the applied Vrec. The larger, longer, and more repetitive was the applied Vrec, the larger was the number of permanent dipoles switched above the coercive field and the greater is the permanent polarization generated, resulting in non-volatile memorization" are simply not true. Or "When Vrec was removed, permanent polarization in the same direction as with Vrec was generated". This cannot happen with a Vrec (like -10V) that is too small to produce hysteresis, for fields much smaller than the coercive field no hysteresis loop opens up and no remnant polarization can be induced.

Response:

We agreed with the comment when the reviewer asked in the previous revision. Therefore, we already fixed and removed the term of "permanent polarization" in the revised manuscript when we carried out the first revision work. We described it in page 13, Comment #5 as follows: "Also, since the SW is originated from the variation of polarization caused by partial switching of dipoles in ferroelectric gate dielectric layer below saturation, we agree with the comment on terminology "permanent polarization" which caused a confusion in meaning. Therefore, we edited the "permanent polarization" to "polarization" or "partial polarization" in the whole manuscript."

8. To sum up, the authors did not show the full hysteresis behaviour of the nanocomposite dielectric, especially not for the interesting voltage range (e.g. +/- 10V) neither did they provide values for the dielectric constant of this material. From the hysteresis curves provided (SI Fig.7, it can be assumed that ferroelectricity plays no role in the proposed AiI-TSO device, since most probably a hysteresis loop will not open up at $V_{rec} = \pm 10V$ (the typical Vrec value that was used for characterization) and no remnant polarization will develop. So the material then behaves rather like an electret or a paraelectric and the observed effects seem to be rather due to the high dielectric constant of the material.

Response:

We appreciate concerns about the ferroelectricity at V_{rec} of +/-10V. As we replied to all previous questions, we checked the P-E curve at +/-10V and other range of voltage to check the minor loops of BT NP(20 wt%)/P(VDF-TrFE) thin film. Although the main voltage band we used for characterization is a voltage below coercivity, it can be applied to generation of synaptic weight by forming partial polarization switching in the minor loop region of ferroelectrics, by controlling various characteristics of electrical or mechanical stimuli (stimulation time, number, speed, magnitude, etc.) for regulating the STP and LTP. This is similar to the principle of researches for verifying synaptic devices using voltages in other minor loop regions and memory devices using ferroelectric material (refs.33-39 in manuscript). Therefore, most synaptic characterizations were performed by setting +/- 10V as the main amplitude of the V_{rec} pulse magnitude. This is more advantageous in terms of linearity and stability when analyzing the synaptic plasticity characterization by applying a voltage higher than the coercivity and is suitable for showing the triboelectric coupling effect. It is considered to be sufficient to prove Fe-OFET as a device that mimics the synaptic function of the sensory organ. Even though we demonstrated the LTP with several hours of retention time as mimicking sensory organ's synaptic functions, it was recognized that applying a voltage higher than the coercive force can form a saturated polarization which provides long and stable retention time. Therefore, we think it is worth studying if we can get a higher and stable long-term memory like using the saturation loop region without unstable linearity in PSC variation, synaptic weight and device breakdown.

Therefore, we added the sentences in discussions section to clarify the needed future works in our research as follows in page 14: “We demonstrated that our AiS-TSO device has adaptation, filtering, and memory functions and shows parallel spatiotemporal reception and preprocessing of tactile information by synaptic weight control using the partial polarization of minor loops in ferroelectric layer. Even though we successfully controlled the synaptic properties which can be considered as mimicking pre-data processing in sensory organ, we can also consider saturation loop of ferroelectric layer to get synaptic properties with longer retention time.”

Reviewer #3

The authors have improved the manuscript. The writing is clearer, helping to understand the work. The comments of the reviewers have been addressed to some extent. Device images provided in supplementary fig. 11 are very interesting.

1. The cross-section image shows an extremely inhomogeneous particle distribution and agglomeration. Large distances between NPs and channel can be observed, raising the question as to how they can affect the channel conductivity.

Response:

We appreciate the thoughtful consideration and advices on previous responses. For the distribution of BT NPs, we checked the SEM image of BT NPs/P(VDF-TrFE) thin film in **Supplementary Figure 22**, and we considered the distribution of BT NPs is quite reasonable compared with our previous reports (Supplementary Information ref. 26,27). For the cross-sectional image in **Supplementary Figure 13**, the distances between layers were caused by delamination of the layer while the device is being cut by FIB. This problem does not occur when the device is intact, and so we would like to say that there is no problem in forming channel though an electric field. If such a problem existed in an uncut device, of course, the device's characteristics would not be obtained. As the results shown, the device worked well as a normal field effect transistor. However, we also decided that we should mention the explanation for clear understanding of this phenomenon.

Therefore, we added the sentence in the caption of **Supplementary Figure 13b** for description of the image as follows: “**The gap between Ni electrode and gate insulating layers was originated from delamination during sample cutting by FIB.**”

2. The concept should be explained and confirmed by a theoretical model of the device (for instance FEM).

Response:

We appreciate the comment about necessity for theoretical analysis of our device. We designed the device concept based on triboelectric effect which has been actively studied for nanogenerators. On our manuscript, we included references on triboelectric pressure sensors (ref. 16,18,60 in manuscript), triboelectric theories and model analysis (ref. 55-57, 59 in manuscript), and these theories which are known. Triboelectric effect between polyimide and skin has also been studied as a representative triboelectric charge layer. (ref.54-57). Therefore, the electrostatic charge transfer between the two materials, the voltage output, etc. are well known, and the results from this study are similar (ref. 54-57) (**Supplementary Figure 19**). This study, which provides the sensor a synaptic function using the triboelectric-capacitive coupling effect, is clearly distinguished from other studies in which the triboelectric sensor is electrically connected with a transistor. However, in so-called tribotronics studies (ref. 57 in manuscript), the theoretical mechanisms in designing this study were publicized. Therefore, there have already been a lot of theoretical analysis related to triboelectric charges exchange or transistor operation with triboelectric energy (ref. 55-57 in manuscript) so that we concluded that theoretical analysis could be sufficiently replaced by other previously analyzed researches. We have done a lot of experiments and have successfully proved the triboelectric-capacitive coupling effect as the main mechanism from other doubtful mechanisms (charge trapping, piezoelectricity, pyroelectricity, or POSFET, **Supplementary Figure 3,17&18**). Therefore, additional theoretical simulation analysis is not essential to explain the important theme here. However, considering reviewer's concerns, we added some references related to triboelectric sensor and theory of triboelectric effect in supplementary information as follows:

Also, we added the sentences to make sure theoretical basis from other research in description in **Supplementary Figure 18** as follows: “Those mechanisms have studied by theoretical analyses on triboelectric effect^{43,50-52} or tribotronics^{43,53,54} although AiS-TSO has uniqueness in intrinsic-synaptic function and structure mimicking MCNCs.”

Here is added references (50-54) related to description above in Supplementary information as follows:

50. Zhang, C. *et al.* Contact electrification field-effect transistor. *ACS Nano*. **8**, 8702–8709 (2014).
51. Shankaregowda, S. A. *et al.* Nano energy single-electrode triboelectric nanogenerator based on economical graphite coated paper for harvesting waste environmental energy. *Nano Energy* **66**, 104141 (2019).
52. Niu, S. & Wang, Z. L. Theoretical systems of triboelectric nanogenerators. *Nano Energy* **14**, 161–192 (2014).
53. Gao, G. *et al.* Triboiontronic transistor of MoS₂. *Adv. Mater.* **31**, 1–10 (2019).
54. Zhang, C. & Wang, Z. L. Tribotronics—A new field by coupling triboelectricity and semiconductor. *Nano Today* **11**, 521–536 (2016).

REVIEWERS' COMMENTS:

Reviewer #2 (Remarks to the Author):

After this second round the reviewer is grosso modo satisfied with the answers explaining the PSC control by partial polarization of the ferroelectric layer and supports the publication of the manuscript.

Therefore the manuscript can be published, after making the following minor corrections:

1. Please change the sentence that was added to the manuscript on page 8 (190-192) according to:

We found that the BT NP(20 wt%)/P(VDF-TrFE) nanocomposite has a smaller coercive field (48.83 MV/m) than that of pure P(VDF-TrFE) (88.2 MV/m), which implies easier polarization switching in the nanocomposite than in P(VDF-TrFE). Nevertheless the coercive field of our P(VDF-TrFE) layer is significantly higher than reported elsewhere. [please cite (1) T. Furukawa in Adv. Colloid Interface Sei. 71-72, 183-208 (1997); and (2) B. Stadlober et al. in Chem. Soc. Rev. 48, 1787 (2019)]

2. Please I still would like to insist that you provide values for the dielectric constant of the materials. On page 7 of the Supporting Information, lines 111-112, where the capacitance values of PVP MIMs and PVDF-TrFe/BT MIMs are mentioned it is highly recommended and inevitable to also indicate the permittivity of the dielectric layers (can easily be deduced from the measured capacitance via the known area and thickness).

RESPONSE TO REFEREES (NCOMMS-19-17002C)

Reviewer #2

1. Please change the sentence that was added to the manuscript on page 8 (190-192) according to:

We found that the BT NP(20 wt%)/P(VDF-TrFE) nanocomposite has a smaller coercive field (48.83 MV/m) than that of pure P(VDF-TrFE) (88.2 MV/m), which implies easier polarization switching in the nanocomposite than in P(VDF-TrFE). Nevertheless the coercive field of our P(VDF-TrFE) layer is significantly higher than reported elsewhere. [please cite (1) T. Furukawa in Adv. Colloid Interface Sei. 71-72, 183-208 (1997); and (2) B. Stadlober et al. in Chem. Soc. Rev. 48, 1787 (2019)

Response:

The comments on coercive field of ferroelectric materials are appreciated. We changed the sentence in manuscript as reviewer recommended as follows (page 8 in manuscript): “**We found that the BT NP(20 wt%)/P(VDF-TrFE) nanocomposite has a smaller coercive field (48.83 MV/m) than that of pure P(VDF-TrFE) (88.2 MV/m), which implies easier polarization switching in the nanocomposite than in P(VDF-TrFE). Nevertheless, the coercive field of our P(VDF-TrFE) layer is significantly higher than reported elsewhere^{41,42}.**”

Also, we added the references 41 and 42 in manuscript as you commented as follows:

41. Furukawa, T. Structure and functional properties of ferroelectric polymers. *Adv. Colloid Interface Sei.* **71-72**, 183-208, (1997).

42. Stadlober, B., Zirkel, M. & Irimia-Vladu, M. Route towards sustainable smart sensors : ferroelectric polyvinylidene fluoride-based materials and their integration in flexible electronics. *Chem. Soc. Rev.* **48**, 1787-1825, (2019).

2. Please I still would like to insist that you provide values for the dielectric constant of the materials.

On page 7 of the Supporting Information, lines 111-112, where the capacitance values of PVP MIMs and PVDF-TrFe/BT MIMs are mentioned it is highly recommended and inevitable to also indicate the permittivity of the dielectric layers (can easily be deduced from the measured capacitance via the known area and thickness).

Response:

The comments on providing the dielectric constants of the materials are appreciated. As you recommended, we added the dielectric constant in explanation of **Supplementary Figure 3** as follows: “The measured capacitance of BT NP(20wt%)/P(VDF-TrFE) was much higher (~ 21 nF/cm² , dielectric constant = 13.81) than PVP (~ 5.2 nF/cm² , dielectric constant = 3.25) in the MIM structure with the same insulator thicknesses as those in the FET structure, which results in the higher on-state current level in the transfer characteristics (**Supplementary Figure 3a**) in Fe-OFET with ferroelectric nanocomposite than that in OFET device with PVP device.”